Shell variability in the stem turtles Proterochersis spp.

http://orcid.org/0000-0001-5108-8493 Szczygielski Tomasz 1 2 t.szczygielski@twarda.pan.pl
Słowiak Justyna 1
Dróżdż Dawid 1
1 Department of Evolutionary Paleobiology, Institute of Paleobiology, Polish Academy of Sciences , Warsaw , Poland
2 Department of Paleobiology and Evolution, Faculty of Biology, Biological and Chemical Research Centre, University of Warsaw , Warsaw , Poland
Anquetin Jérémy
Electronic publication date: 2018 Dec 21
Publication date: 2018
Volume: 6
Electronic Location ID: e6134
Received 2018 Jun 27; Accepted 2018 Nov 19
Copyright: © 2018 Szczygielski et al.
Copyright year: 2018
Copyright holder: Szczygielski et al.
License: This is an open access article distributed under the terms of the Creative Commons Attribution License, which permits unrestricted use, distribution, reproduction and adaptation in any medium and for any purpose provided that it is properly attributed. For attribution, the original author(s), title, publication source (PeerJ) and either DOI or URL of the article must be cited.
License URL: https://creativecommons.org/licenses/by/4.0/

Keywords: Testudinata, Triassic, Turtles, Proterochersidae, Reptilia, Mesozoic, Shell, Carapace, Plastron, Scutes

Funding: National Science Centre (Narodowe Centrum Nauki), Poland 2016/23/N/NZ8/01823 This work was supported by the National Science Centre (Narodowe Centrum Nauki), Poland, grant no. 2016/23/N/NZ8/01823. The funders had no role in study design, data collection and analysis, decision to publish, or preparation of the manuscript.

==============================
Background

Turtle shells tend to exhibit frequent and substantial variability, both in bone and scute layout. Aside from secondary changes, caused by diseases, parasites, and trauma, this variability appears to be inherent and result from stochastic or externally induced flaws of developmental programs. It is, thus, expected to be present in fossil turtle species at least as prominently, as in modern populations. Descriptions of variability and ontogeny are, however, rare for fossil turtles, mainly due to rarity, incompleteness, damage, and post-mortem deformation of their remains. This paper is an attempt at description and interpretation of external shell variability in representatives of the oldest true turtles, Proterochersis robusta and Proterochersis porebensis (Proterochersidae, the sister group to all other known testudinatans) from the Late Triassic (Norian) of Germany and Poland.

Methods

All the available shell remains of Proterochersis robusta (13 specimens) and Proterochersis porebensis (275 specimens) were studied morphologically in order to identify any ontogenetic changes, intraspecific variability, sexual dimorphism, and shell abnormalities. To test the inferred sexual dimorphism, shape analyses were performed for two regions (gular and anal) of the plastron.

Results

Proterochersis spp. exhibits large shell variability, and at least some of the observed changes seem to be correlated with ontogeny (growth of gulars, extragulars, caudals, and marginals, disappearance of middorsal keel on the carapace). Several specimens show abnormal layout of scute sulci, several others unusual morphologies of vertebral scute areas, one has an additional pair of plastral scutes, and one extraordinarily pronounced, likely pathological, growth rings on the carapace. Both species are represented in a wide spectrum of sizes, from hatchlings to old, mature individuals. The largest fragmentary specimens of Proterochersis porebensis allow estimation of its maximal carapace length at approximately 80 cm, while Proterochersis robusta appears to have reached lower maximal sizes.

Discussion

This is the second contribution describing variability among numerous specimens of Triassic turtles, and the first to show evidence of unambiguous shell abnormalities. Presented data supplement the sparse knowledge of shell scute development in the earliest turtles and suggest that at least some aspects of the developmental programs governing scute development were already similar in the Late Triassic to these of modern forms.

Introduction

The shell of turtles, although relatively conserved structurally among taxa, tends to show considerable variation between individuals (Parker, 1901; Gadow, 1905; Newman, 1906a; Coker, 1910; Lynn, 1937; Młynarski, 1956; Zangerl & Johnson, 1957; Zangerl, 1969; McEwan, 1982; Rothschild, Schultze & Pellegrini, 2013; Cherepanov, 2015, 2016; Farke & Distler, 2015; and many others). This variation may be potentially caused by numerous factors, out of which a suboptimal humidity (Lynn & Ullrich, 1950) or temperature (Yntema, 1970) during incubation, and a low genetic variation (bottleneck) within populations (Velo-Antón, Becker & Cordero-Rivera, 2011; McKnight & Ligon, 2014) were proposed. Expressions of atavistic morphologies were frequently cited as a cause of abnormal shell variants (Gadow, 1905; Newman, 1906b; Grant, 1936a, 1936b), but this always remained rather speculative (Coker, 1905, 1910; Cherepanov, 1989, 2006, 2014) and in most cases it is easy to refute by comparison with the shell composition (number and layout of shell elements) of stem turtles (Gaffney, 1990; Li et al., 2008; Szczygielski & Sulej, 2016, 2018). In some cases, abnormal morphologies are attained during postnatal life as a result of diseases, parasites, or trauma (Rothschild, Schultze & Pellegrini, 2013, and references therein).

Shell variation affects both the bones and scutes of the plastron and carapace, and the frequency of changes within each of these domains varies between the species (Coker, 1910; Lynn, 1937; Zangerl & Johnson, 1957; Zangerl, 1969; McEwan, 1982) and may even differ between sexes within one species (Coker, 1910). Among modern turtles, Cheloniidae are known to have especially variable shells (Kordikova, 2002; Özdemir & Türkozan, 2006; Pritchard, 2008), although large (yet not documented in the literature) variability may also be observed in some pleurodires (G. Ferreira, 2018, personal communication). This unequal susceptibility of various turtles, even those inhabiting similar environments, suggests the presence of some control or repair mechanisms that limit appearance of abnormal morphologies with varying efficiency in different taxa or sexes, but the exact molecular or morphogenetic background of these mechanisms is little known. The developmental rules governing the appearance of supernumerary or asymmetric scutes, however, are well explained by recent studies (Cherepanov, 1989, 2006, 2014, 2015; Moustakas-Verho et al., 2014; Moustakas-Verho & Cherepanov, 2015; Moustakas-Verho, Cebra-Thomas & Gilbert, 2017). According to those studies, shell scutes originate from placodes, which develop in strict correlation with body segmentation: lack of placodes, their asymmetry, improper fusion, or appearance of additional placodes on the level of vacant myosepta lead to abnormal (usually asymmetrical) development of scutes. Some scutes (usually cervical and vertebrals) develop from fusion of initially separate, paired placodes. Some developmental information may, therefore, be obtained from the layout of scutes relative to each other (e.g., see Szczygielski, 2017, for discussion on scutation of Triassic turtles) and even from some scute abnormalities. Understanding of scute development is crucial, because shell scutes precede shell bones in development and thus have a large impact on the external morphology and even layout of the shell bones (Zangerl, 1939, 1969; Cherepanov, 1989, 2006, 2016).

Various congenital changes to the typical shell structure differ in severity. Cherepanov (2016) classified them into three main categories: malformations (severe developmental flaws, usually lethal or severely detrimental), anomalies (changes to the number and layout of shell elements, not severely detrimental, possibly adaptive), and individual variation (minor changes to the number and layout of shell elements, neutral to normal function). Based on this classification, anomalies and individual variations are much more common than malformations and, out of the former two, anomalies are generally easier to spot and understand in the fossil record, because they are usually more pronounced, frequently asymmetrical, and easier to differentiate from post-mortem deformation.

Turtle shells preserve relatively easily in the fossil record, but still, many extinct turtle taxa are known from relatively few, incomplete and/or distorted specimens. For that reason, descriptions of their variability and ontogeny are rare, especially for Mesozoic forms (Gaffney, 1990; Lichtig & Lucas, 2017; Sullivan & Joyce, 2017). Here, we describe the external variability and abnormalities observed in the carapace and plastron of Proterochersis robusta Fraas, 1913 and Proterochersis porebensis Szczygielski & Sulej, 2016 (Figs. 1–10; Figs. S1–S8)—representatives of the earliest branch of true (total-group) turtles (Testudinata) (Szczygielski & Sulej, 2016).

Figure 1 Nomenclature of turtle scutes shown on the reconstruction of the shell of Proterochersis robusta in (A) dorsal, (B) lateral left, and (C) ventral view, and the legend of color and pattern codes used.

Materials and Methods

Proterochersis robusta

Proterochersis robusta (Figs. 1–3, 6, 8, 9A–9C, 10; Figs. S5A–S5C, S6A, S7A–S7D, S8A–S8D; Articles S2–S3; Tables S1 and S2) is known from the Late Triassic (middle Norian) Löwenstein Formation, Stuttgart proximities, Germany. For the geological background, see Szczygielski & Sulej (2016) and references therein. All of the existing specimens of that species (SMNS 11396, SMNS 12777, SMNS 16442, SMNS 16603, SMNS 17561, SMNS 17755, SMNS 17755a, SMNS 17756, SMNS 17930, SMNS 18440, SMNS 50917, SMNS 51441, SMNS 56606, SMNS 81917; CSMM uncat.) were studied with exception of SMNS 50918 (an internal mold of the shell). For the detailed description of these specimens see Article S1 and for the chart of scute areas preserved on each of them see Tables S1 and S2.

Proterochersis porebensis

Proterochersis porebensis (Figs. 4, 5, 7, 8, 9D–9T, 10; Figs. S1, S4, S5D–S5N, S6B–S6D, S7E–S7P′, S8E–S8F; Articles S2 and S3; Tables S3 and S4) is known from the lower part of Patoka Member of Grabowa Formation, Poręba, Poland. For geological and paleoenvironmental background, see Sulej, Niedźwiedzki & Bronowicz (2012), Niedźwiedzki et al. (2014), Szulc et al. (2015), Zatoń et al. (2015) and Szczygielski & Sulej (2016). All of the existing specimens (ZPAL V.39/1–28, ZPAL V.39/34, ZPAL V.39/48–72, ZPAL V.39/155–300, ZPAL V.39/331–366, ZPAL V.39/370, ZPAL V.39/373–404, ZPAL V.39/416–420, and uncataloged) were studied. For the detailed description of these specimens see Article S1 and for the chart of scute areas preserved on each of them see Tables S3 and S4.

Methods

The macrophotographs of the smallest specimens, ZPAL V.39/381 and ZPAL V.39/384, were taken using Keyence Digital Microscope VHX-900F. The terminology used for the shell elements (Fig. 1) follows Zangerl (1969) with the amendment by Hutchison & Bramble (1981). To avoid confusion, we use terms “external” rather than “dorsal,” “lateral,” or (in case of plastron) “ventral” to describe the scute-covered surfaces of the shell, and “dorsomedial” rather than “dorsal” or “medial” when referring to the parts of the carapacial scute areas located closest to the neural row (at the midline and at the very top of the carapace), with the exception of the cervical and the vertebrals, for which the term “medial” is uncontroversial, and the bridge marginals in ventral aspect, for which “ventromedial” is used to indicate the direction toward the middle point of plastron. Also for clarity, for marginal scutes we use “length” for the dimension of marginal scutes measured radially from the middle to the periphery of the carapace, and “width” for the dimension measured along the edge of the carapace, regardless of the position of the scute and thus its orientation relative to craniocaudal axis of the whole carapace. The edge of the marginal scute contacting scutes other than the preceding or succeeding marginal is called “base,” while its free edge is called “rim.”

Data acquisition and preparation

The shape analysis was performed for the gular and anal regions of the plastron. The sample of gular regions includes ten specimens of Proterochersis porebensis (ZPAL V.39/34, ZPAL V.39/48, ZPAL V.39/49, ZPAL V.39/187, ZPAL V.39/333, ZPAL V.39/379, ZPAL V.39/385, ZPAL V.39/387, ZPAL V.39/388, and ZPAL V.39/420) and one Proterochersis robusta individual (SMNS 17561). We took photographs of articulated and most complete cranial tips of plastra (areas of gular and extragular scutes) in ventral view. Additionally the shape of the extragulars of Proterochersis porebensis was analyzed based on the vertical sections of their 3D models. Models were generated photogrammetrically using the freeware program Visual SFM 0.5.26 and the free software MeshLab 2016 (Cignoni et al., 2008; Wu, 2011) for the specimens ZPAL V.39/48, ZPAL V.39/187, ZPAL V.39/333, ZPAL V.39/379, ZPAL V.39/385, ZPAL V.39/387, ZPAL V.39/388, and ZPAL V.39/420 and Agisoft Photoscan 1.2.0 (www.agisoft.com) for ZPAL V.39/34 and ZPAL V.39/49. In the case of the Proterochersis robusta specimen SMNS 17561 (the only specimen of that species fully preserving that region) the 3D modeling was impossible to perform because the specimen is not sufficiently prepared and the extragulars are only exposed ventrally. To create the models with Visual SFM and MeshLAB, around 100 photos from angles of 0°, 15°, and 40° were taken. To speed up the process of the model generation, the photographs were scaled down to resolution 800 × 600 px by the freeware program FastStone Photo Resizer 3.8 (www.faststone.org) with the exception of ZPAL V.39/385, for which photographs of resolution 6,000 × 4,000 px were used. For the specimens ZPAL V.39/34 and ZPAL V.39/49, around 200 of 6,000 × 4,000 px photos were used to obtain models of whole shells, and then the cranial plastral sections were separated using MeshLab. Model of each specimen was then scaled in MeshLab to match its original size and the extragulars were digitally sectioned in sagittal plane. For sectioning the best preserved extragulars were chosen: right extragulars of ZPAL V.39/49, ZPAL V.39/187, ZPAL V.39/333, ZPAL V.39/379, ZPAL V.39/388, left extragulars of ZPAL V. 39/34, ZPAL V.39/387, ZPAL V.39/420 right and left extragulars of ZPAL V.39/48 and ZPAL V.39/387. The 3D models used for the analysis are shown in the online appendices Articles S2 and S3.

For the anal region, we took photographs of the best preserved caudal tips of plastra (caudal processes and intercaudal scute) in ventral view. We analyzed seven Proterochersis porebensis (ZPAL V.39/34, ZPAL V.39/48, ZPAL V.39/49, ZPAL V.39/66, ZPAL V.39/69, ZPAL V.39/70, and ZPAL V.39/71) and three Proterochersis robusta individuals (CSMM uncat., SMNS 12777, and SMNS 17561). The photographs were converted to the .TPS format using the tpsUtil 1.76 program (Rohlf, 2015).

A set of five landmarks on the gular region in ventral view, two in the vertical section of the extragulars, and another five on the anal region were digitalized using the software tpsDig 2.31 (Rohlf, 2015). All landmarks are of Bookstein’s (1997) first type, they point the intersections between element boundaries—herein, the landmarks mark intersections between the sulci (see Table 1). In order to analyze the shape of curves, we used eight semilandmarks along the cranial curve of the gular and extragular, 13 along the curve of cross-sectioned extragular, and 12 semilandmarks along the edge of caudal process. The semilandmarks were also taken using tpsDig. In case of minor incompleteness of the element, several semilandmarks were estimated, using the opposite side whenever possible. In four cases (SMNS 17561, ZPAL V.39/49, ZPAL V.39/69, and ZPAL V.39/385), complete data from opposite sides of a single individual were averaged. The gular and the anal regions were digitized twice.

Table 1 Definition of landmarks used in principal component analyses.

Plastron region	Plane	Landmark	Definition	
Gular	Ventral	1	Caudal end of the intergular sulcus	
2	Caudal end of the sulcus between the gular and extragular scute	
3	Laterocaudal tip of the extragular scute	
12	Cranial end of the sulcus between the gular and extragular scute	
21	Cranial end of the intergular sulcus	
Vertical cross-section	1	Sulcus between the extragular and humeral scute	
15	Posterodorsal edge of the extragular scute	
Anal	Ventral	1	Cranial tip of the intercaudal scute	
2	Caudal edge of the intercaudal scute	
3	Cranial end of the caudointercaudal sulcus	
4	Caudal end of the caudointercaudal sulcus	
17	Lateral end of the caudoanal sulcus	

Geometric morphometric analyses

All geometric morphometric analyses were performed in MorphoJ 1.06d (Klingenberg, 2009, 2011). First, generalized Procrustes analysis was performed to remove the information related to size, position, and orientation (Zelditch et al., 2004). Then, we averaged the shape data for left and right sides for ZPAL V.39/49, ZPAL V.39/69, and ZPAL V.39/385. Wireframe graphs and principal component analyses (PCA) were used to visualize shape differences of gulars, extragulars, and caudal processes. Significance of the differences in shape between the groups identified form the PCA plots were assessed using the Procrustes Analysis of Variance, analogous to multivariate analysis of variance (MANOVA) (Klingenberg, 2009, 2011). A multivariate regression of the caudal processes and gular shape against log-transformed centroid size on pooled within-group (by species) variation was conducted for the presence of allometries (Klingenberg, 1996).

The Pearson product-moment correlation coefficients used for simple estimations of the shell sizes of exceptionally large, fragmentary specimens were calculated in Microsoft Office Standard 2010 Excel (v. 14.0.7212.5000, 64-bit) using the CORREL function. These calculations are based on very small sample sizes, therefore they should be understood as auxiliary only.

Results

Specimen sizes

Shell material of Proterochersis robusta consists of 13 specimens of varying sizes and, supposedly (inferred on size and morphology), ontogenetic age spanning from a young, not yet fully ossified juvenile (SMNS 81917, Fig. S6A) to large, apparently mature, individuals (e.g., SMNS 16442, Figs. 2C, 3D, or SMNS 18440, Figs. 2K and 3H). Shell remains of Proterochersis porebensis are much more numerous (270 cataloged specimens), but usually much more fragmentary, frequently consisting of parts of costals, small sections of plastron or the rim of the shell, or other uninformative elements, and only four relatively complete shells (ZPAL V.39/34, ZPAL V.39/48, ZPAL V.39/49, and ZPAL V.39/72) were found thus far (Figs. 4 and 5).

Figure 2 External carapace morphology of Proterochersis robusta.

(A and B) CSMM uncat. in (A) dorsal and (B) lateral right view; (C) SMNS 16442, carapace in dorsal view; (D and E) SMNS 16603 in (D) dorsal and (E) lateral right view; (F and G) SMNS 17561 in (F) dorsal and (G) lateral left (mirrored) view; (H) SMNS 17755a in dorsal view; (I and J) 17930 in (I) dorsal and (J) lateral right view. (K) SMNS 18440 in lateral left (mirrored) view. Restored area not shown for SMNS 17561 (F and G) due to difficulties in evaluation. Minor damage and restorations not shown for clarity.

Figure 3 External plastron morphology of Proterochersis robusta.

(A) CSMM uncat. in ventral view; (B) SMNS 11396, plastron in ventral view; (C) SMNS 12777 in ventral view; (D) SMNS 16442, plastron in ventral view; (E) SMNS 16603, plastron in ventral view; (F) SMNS 17561 in ventral view; (G) SMNS 17755, plastron in ventral view; (H) SMNS 18440 in ventral view; (I) SMNS 50917 in ventral view; (J) SMNS 56606 in ventral view. Scute sulci are represented by dashed gray lines. Restored area not shown for SMNS 17561 (F) due to difficulties in evaluation. Minor damage and restorations not shown for clarity.

Figure 4 External carapace morphology of Proterochersis porebensis.

(A and B) ZPAL V.39/34 in (A) dorsal and (B) lateral left (mirrored) view; (C and D) ZPAL V.39/48, (C) carapace in dorsal and (D) lateral right view; (E and F) ZPAL V.39/49, (E) carapace in dorsal and (F) lateral right view; (G and H) ZPAL V.39/72 in (G) dorsal and (H) lateral left (mirrored) view. Minor damage and restorations not shown for clarity.

Figure 5 External plastron morphology of Proterochersis porebensis.

(A) ZPAL V.39/34 in ventral view; (B) ZPAL V.39/48 in ventral view; (C) ZPAL V.39/49 in ventral view. Minor damage and restorations not shown for clarity.

Similarly to Proterochersis robusta, the collected specimens of Proterochersis porebensis represent a wide spectrum of sizes and morphologies, probably representing various ontogenetic ages. The smallest known individual appears to be a hatchling or a very young juvenile, and is represented by an exceptionally small, fragmentary costal (ZPAL V.39/381, Figs. S1C and S1D). ZPAL V.39/34 (Figs. 4A, 4B, 5A and 9K; Figs. S4C, S5G and S7E–S7G) is more developed and much larger (approx. 28 cm of carapace length; note that shell lengths are approximate due to damage and distortion), but still differs morphologically from the larger specimen and thus may be tentatively classified as an older juvenile. ZPAL V.39/48 (Figs. 4C, 4D, 5B and 9G; Figs. S2A, S2B, S5H and S7H–S7J) is even larger (approx. 42.5 cm of carapace length) and exhibits shell morphology typical for large individuals but is tentatively classified as a subadult based on incompletely ossified dorsal process of the scapula (following Gaffney, 1990; see Szczygielski & Sulej, 2016). ZPAL V.39/72 (Figs. 4G and 4H; Figs. S4D and S5K) is of comparable size (approx. 44.5 cm of carapace length) and seems to be of comparable ontogenetic age. ZPAL V.39/49 (Figs. 4E, 4F, 5C and 9D; Figs. S2C–S2F, S5I and S7K–S7M) is the largest complete shell found thus far (approx. 49 cm of carapace length) and thus is treated as an adult, but some fragmentary specimens, such as ZPAL V.39/8 (see Szczygielski & Sulej, 2018), ZPAL V.39/57 (Figs. S1N and S3B), ZPAL V.39/60 (Figs. S1O and S1P), and ZPAL V.39/63 (Figs. S1A and S1B) indicate that this species could have reached even larger sizes. ZPAL V.39/63 (a carapace fragment with dorsal part of ilium attached) seems to be particularly large—around the contact with the ilium, the carapace is up to 1.5 cm thick, the sulci are very wide (see below), and the ilium is massive, being at the point of attachment to the carapace 6.3 cm broad (measured from the lateral edge of the bulge to the base of the first sacral rib), compared to 3.5 cm in ZPAL V.39/48, four cm in ZPAL V.39/49, and 3.7 cm in ZPAL V.39/72. Accurate measurement of ilium breadth is difficult in ZPAL V.39/34 due to damage and surrounding rock matrix, but it seems to be about two cm. Based on these data, it seems that there is a strong correlation between the breadth of the dorsal most end of the ilium and the length of the carapace (Pearson correlation coefficient = 0.997 with ZPAL V.39/34 included and 0.995 with that specimen excluded; n = 4 or n = 3, respectively), although the small number of specimens must be taken into account. Based on that, the carapace length of ZPAL V.39/63 may be estimated to be between 75 and 80 cm (depending on whether the measurement of ZPAL V.39/34 is considered). With the exception of ZPAL V.39/34, the collected complete shells of Proterochersis porebensis are larger than all known shells of Proterochersis robusta (possibly excluding the fragmentary specimens SMNS 16442 and SMNS 18440, as their exact size is difficult to estimate).

It must be kept in mind that in most cases it is very difficult or outright impossible to accurately estimate the ontogenetic age or even general ontogenetic stage of fossil animals, especially when the specimens are fragmentary and no skeletochronological data are at hand. Usually, the only readily available proxy for the relative age of the individuals is their size and, in some cases, advancement of the skeletogenesis (ossification of the articular surfaces of long bones, closure of fontanelles, fusion of sutures, etc.). The sizes and tempo of ossification in the animals of the same age may, however, differ within the population depending, for example, on their sex or environmental conditions (Berry & Shine, 1980; Pritchard, 2008; Farke & Distler, 2015). For that reason, animals of different sizes may be of comparable ontogenetic age and animals of similar size may in fact represent different age groups. Based on some consistent combinations of size and morphological characteristics, in the studied material of Proterochersis spp. several ontogenetic stages may be provisionally recognized—most notably supposed juveniles (small, sometimes not fully ossified shells, weakly developed gular and extragular tubercles and caudal processes, straight margins of the shell, etc.), adults (medium-sized shells, prominent gular and extragular tubercles, well-developed caudal processes and features of the shell margin, etc.), and old individuals (largest shells, most pronounced superficial characteristics of the shell; see below for detailed characterization). For convenience, these categories will be evoked in the descriptions, but they are of course tentative, given the aforementioned problems, and not entirely discrete. For now, they should be treated as generalized morphotypes with implied ontogenetic meaning. Obviously, it is unknown at which point these animals attained sexual maturity, so our classification into the juveniles and adults is based solely on morphological terms, although it seems possible that at least some of the adult-like characters (particularly the pronounced gulars, extragulars, and caudals) may be related to sexual selection or copulation. Skeletochronology will probably be attempted for Proterochersis porebensis in the future and these categories will possibly be validated.

There is some incongruency between these large maximal sizes and the moment of shell ankylosis. Typically, the ankylosis stops the growth of the shell, because the bones grow mainly along the sutures (Pritchard, 2008). Most specimens of Proterochersis spp., however, are fully ankylosed, regardless of their size. Even if the prevalence of ankylosed specimens in Poręba and localities around Stuttgart may be a preservation or sorting artifact (e.g., the unankylosed specimens were typically destroyed by currents or small fragments of disarticulated unankylosed shells were buried elsewhere; see Szczygielski & Sulej, 2018), the fact that ankylosis occurred even in relatively small specimens with juvenile features (e.g., ZPAL V.39/2, ZPAL V.39/34, ZPAL V.39/66) is more troubling. Many of these small specimens are well-preserved and it is hard to imagine that the sutures were obliterated by some diagenetic processes, while minor details of shell anatomy and texture remained intact. In some turtle species the sexual dimorphism takes form of a striking difference of sizes between mature males and females (Pritchard, 2008). In such a case, specimens like ZPAL V.39/34 could be considered one of the sexes, and large specimens like ZPAL V.39/49—the other one. This, however, seems to be refuted by a fact that there exists a full spectrum of sizes of ankylosed specimens between ZPAL V.39/34 and ZPAL V.39/49 (e.g., ZPAL V.39/48 with subadult characters). Likewise, this would preclude interpretation of small ankylosed specimens as a separate species. In lack of other likely explanations, a very broad variation in time of ankylosis is therefore provisionally accepted. Another possible solution is seasonal hypercalcifiaction and decalcification of sutures or shell bones that could increase the rigidness of the shell mosaic (see Szczygielski & Sulej, 2018) but also allow seasonal growth—similar mechanism of de-ossification was reported locally in the mid-section of plastron in males of some modern turtles during mating season (Wibbels, Owens & Rostal, 1991; Wyneken, 2001; Pritchard, 2008). Finally, the growth might have occurred mainly by the means of bone remodeling. This problem may be resolved by future histological studies.

With very few exceptions, the specimens of Proterochersis spp. are incomplete, and often the overlap between them is small, which makes comparisons or even reliable estimation of their size difficult—even more so, relative proportions of epidermal elements or breadth of plastral lobes may vary between individuals and sometimes even bilaterally within one individual, as evidenced by several relatively complete shells. For that reason, it is impossible to confidently estimate the shell length based on, for example, the length of a single plastral scute. These differences in proportions are difficult to explain, and incompleteness or poor preservation of the specimens makes it currently impossible to determine if they result, for example, from allometric growth, sexual dimorphism, or are just part of normal intraspecific variability.

See Table S5 for the collection of most important measurements of Proterochersis spp. given in the text.

Carapace

Costals

ZPAL V.39/381 (Figs. S1C and S1D) is a fragmentary costal of the smallest, and probably the youngest, known individual of Proterochersis porebensis. This costal is eight mm wide, two mm thick in the peripheral regions, and three mm thick at the ventral ridge. It appears to lack natural edges with exception of a portion of proximal (medial) rim. The gracility of that element, its smooth external surface with subtle longitudinal striation, lack of the typical rough texture indicative of contact with superficial layers of dermis, and a rounded convexity in the proximal region of the external surface (Fig. S1C) suggest, however, that it is not a part of a full-sized costal.

The widest costal with preserved sutural edges is ZPAL V.39/176 (5.1 cm wide, 5 mm–1.2 cm thick, Fig. S1E). Its width suggests that it comes from an individual similar in size to ZPAL V.39/49. The structure of the sutures is relatively simple (longitudinally lamellar) in that specimen and the thickness is intermediate, compared to some other specimens (see Szczygielski & Sulej, 2018), even if they are narrower. This probably results from the position of the costal within the shell—as a general rule, the posterior costals seem to be narrower in Proterochersis spp. than the anterior ones. Thus, it is likely that the thicker costals with more developed sutural edges, such as ZPAL V.39/3 (see Szczygielski & Sulej, 2018) come from ontogenetically older specimens, but from more caudal part of the shell.

Vertebrae

ZPAL V.39/377 (Figs. S1H–S1K and S2I) is a fragment of the dorsal section of the vertebral column of a young Proterochersis porebensis individual, consisting of one and a half vertebra. Besides the relatively small size (the complete vertebra is 1.9 cm long, 1.2 cm wide at the level of facets for the ribs), it differs from all other known vertebrae of Proterochersis spp. in lack of ankylosis and neurals. The natural bone limits are visible, proving that the dorsal ribs in proterochersids were already shifted cranially to an intervertebral position typical for turtles. The facets for the ribs (Fig. S1I) are ovoid, longer than high, higher caudally than cranially, gently skewed cranioventrally in lateral view, and at least in ¾ of their length they are located in the cranial part of the centrum. Likewise, the neural spines are also inclined slightly cranially. The neurocentral sutures cross the articulation facets for ribs, their inclination is slightly oblique, dorsocaudal, and generally agrees with the inclination of the facets. The zygapophyses are small and roughly triangular. The centra are hourglass-shaped in ventral view (Fig. S1J). Along the long axis of the centra, their ventral surfaces are gently projected ventrally. As they are preserved, the vertebrae are separated by a gap approx. three mm wide (Figs. S1I and S1J), probably filled in life by the intervertebral disc or unossified, cartilaginous ends of the centra. The neural canal, exposed caudally, is very high and narrow, measuring up to eight mm in height and 2.5 mm in width (Fig. S1H). The most surprising is the dorsal surface of the neural spines (Fig. S1K). Neurals are absent, but there is no sign of bone breaking, and no cancellous bone is exposed. Instead, the surface is bumpy and perforated by numerous vascular canals. This does not resemble a suture; there are no lamellae nor spikes. For that reason, we interpret this either as a sign of a cartilaginous cap on the dorsal ends of neural spikes (albeit it is located relatively high and the neural spikes are broadened dorsally, Fig. S1H) or as incipient intramembranous ossification, just beginning the development of neurals. In either case, it indicates young ontogenetic age of the individual.

All the other specimens of Proterochersis spp. that preserve dorsal vertebrae, including SMNS 56606, ZPAL V.39/48 (see Szczygielski & Sulej, 2016; Szczygielski, 2017), ZPAL V.39/49 (see Szczygielski, 2017), ZPAL V.39/72 (see Szczygielski & Sulej, 2016; Szczygielski, 2017), ZPAL V.39/169 (Figs. S2G, S4E and S4F, comparable in size with ZPAL V.39/49), and ZPAL V.39/378 (Figs. S1F, S1G and S2J comparable in size with ZPAL V.39/49) have their dorsal vertebral columns fully ankylosed, and no unambiguous intervertebral and costovertebral articulation points or sutures can be seen. In these larger (and supposedly ontogenetically older) specimens the dorsal vertebrae get obviously larger and more robust, particularly at the points of articulation. The ventral surfaces of the dorsal vertebrae of ZPAL V.39/48 (with exception of the first and the last three dorsal vertebra) form a relatively sharp midventral ridge (Fig. S2B; see Szczygielski & Sulej, 2016), and a sharpened ridge can be seen on the mid-dorsal vertebra of ZPAL V.39/49 (Figs. S2C and S2D), but in the remaining specimens the ventral surface is more rounded. Given the limited sample, it is difficult to tell if this is related to ontogeny or just variable in the population. It seems that this sharpened ridge is more frequent in the mid-section of the dorsum than in the cranial or caudal end of the dorsal vertebral column (Fig. S2). The dorsal neural canal of ZPAL V.39/49 (Figs. S2C and S2E), ZPAL V.39/169 (Fig. S2G), ZPAL V.39/378 (Fig. S2J), and ZPAL V.39/402 (Fig. S2K), as exposed, is closer to circular in cross-section and measures approx. 4 × 5 mm in ZPAL V.39/169 and ZPAL V.39/378, approx. 8 × 7 mm in ZPAL V.39/49, and approx. 8 mm × 1 cm in ZPAL V.39/402. The cranial opening of the dorsal canal at the cranial end of the first dorsal vertebra in ZPAL V.39/48 is more subtriangular or ovoid and measures approx. 7 × 6 mm (Fig. S2A). The observations made during the preparation of ZPAL V.39/49 indicate that the shape of the canal changes craniocaudally (more oval in the parts with the sharp ventral ridge, more circular caudally; no exact measurement were taken at the time; Figs. S2C–S2F), but never seem to approach the proportions seen in ZPAL V.39/377. Similarly, the cross-section of the mid-dorsal vertebral column exposed during preparation in ZPAL V.39/48 (Fig. S2B) reveals a circular neural canal (also not measured, but the width to height ratio approached 1:1). ZPAL V.39/370 shows a circular neural canal (8 × 7 mm) in the cross-section of the first sacral vertebra (Fig. S2H) and ZPAL V.39/402 shows a triangular neural canal (1 cm × 9 mm) the end of the second sacral vertebra (both co-ossified with the carapace; Fig. S2L). Although compaction might have potentially had some impact on the shape of the vertebral canal in mentioned specimens, no significant deformation is apparent.

Cervical scute

In adult and subadult individuals of Proterochersis robusta and Proterochersis porebensis the cervical was trapezoidal to crescent-shaped (Figs. 1, 2, 4, 6B and 6C; Figs. S1N and S3). The caudal (basal) edge, contacting the cranialedge of the first vertebral scute, was longer than the cranial. The shortest, slanted, craniolateral edges contacted the mediocaudal edges of the first pair of marginal scutes. The lateral tip of the cervical scute may form a several millimeter long contact with the base of the second marginal scute (e.g., Proterochersis porebensis specimens ZPAL V.39/57, Figs. S1N, S3B, and ZPAL V.39/49, Fig. 4E), merely touch the second marginal (e.g., Proterochersis porebensis ZPAL V.39/48, Fig. 4C, and ZPAL V.39/72, Fig. 4G), or such a contact may be prevented by the first marginal (e.g., Proterochersis robusta SMNS 17561, Fig. 2F, and SMNS 17930, Figs. 2I, 6B and 6C; Proterochersis porebensis ZPAL V.39/22, Fig. S3A)—these morphologies seem to not be discrete, but rather form a continuous series of variable relative positions of the lateral tip of the cervical and the sulcus between the first two marginals. In Proterochersis porebensis specimen ZPAL V.39/34 (Fig. 4A) the cervical was more rectangular, with roughly craniocaudally directed lateral edges. In ZPAL V.39/34 (Fig. 4A) the cervical is eight mm long, in ZPAL V.39/390 (Fig. S1L) it is one cm long, in ZPAL V.39/22 (Fig. S3A) and ZPAL V.39/48 (Fig. 4C) it measures 1.5 cm in length, in ZPAL V.39/72 (Fig. 4G) it is 1.9 cm long, and in ZPAL V.39/49 (Fig. 4E) it is 2.2 cm long. ZPAL V.39/57 (Fig. S1N) has the longest cervical, measuring 2.4 cm. In most specimens the cervical scute breadth did not exceed 1/3 of the width of the first vertebral scute, but in ZPAL V.39/49 the cervical is wider than a half of the first vertebral (Fig. 4E). It remains uncertain whether this is correlated with the presence of paired nuchal bone (see Szczygielski & Sulej, 2018) or if it results, for example, from allometry.

Vertebral scutes

Proterochersis spp. had a single row of five broad vertebral scutes, which covered most of the dorsal surface of the carapace (Figs. 1, 2, 4 and 6). The first vertebral was fan-shaped, with a rounded medial process directed caudally, which was received by a cranial medial notch of the second vertebral. The cranial edge was gently bowed, it contacted the caudal (basal) edge of the cervical, the base of the second marginal, and (usually) cranial portion of the base of the third marginal (Proterochersis porebensis ZPAL V.39/57, Figs. S1N and S3B, being the only known exception due to the second marginal preventing such contact). In some specimens (e.g., Proterochersis robusta specimens SMNS 17561, Fig. 2F, and SMNS 17930, Figs. 2I, 6B and 6C, and Proterochersis porebensis ZPAL V.39/22, Fig. S3A) there is a minor contact between the first vertebral and the caudal portion of the base of the first marginal scute. Laterally, the first vertebral formed facets for contact with the first pair of pleurals. The length of these facets proportionally to the scute area increased with the size of the animal, in large individuals (such as Proterochersis porebensis ZPAL V.39/49, Fig. 4E, and ZPAL V.39/57, Fig. S1N) reaching over 3.5 cm. The first vertebral in some specimens was slightly asymmetrical—in SMNS 17561 its left caudolateral margin was more concave than the right one (Fig. 2F), in ZPAL V.39/49 the scute was expanded slightly more to the right than to the left (Fig. 4E), and in ZPAL V.39/72 the caudalmost point of the caudal process seems to be shifted to the left relative to the midline (Fig. 4G). These changes do not seem to result from the crushing or compaction of the specimens.

The cranial edge of the second vertebral was bow-shaped, with a rounded medial embayment which received the caudal process of the first vertebral scute. Craniolaterally, it contacted the dorsomedial edges of the first pair of pleurals, laterally it contacted about 3/5 of the dorsomedial edge of the second pair of pleurals, and caudally it contacted the cranial edge of the third vertebral scute. The second vertebral is widest in its caudal part, at (or slightly caudal to) the level of the sulcus between the first and the second pleural. The remaining vertebrals were roughly trapezoidal, each scute slightly narrower caudally than cranially, and had generally straight cranial edges. The third vertebral contacted the second vertebral cranially, the remaining part of the dorsomedial edge of the second pair of pleurals and over 2/3 of the dorsomedial edge of the third pair of pleurals laterally, and the fourth vertebral scute caudally. It was widest around the sulcus between the respective pleurals, and in dorsal view its width was comparable to the width of the second vertebral (although it might have been slightly larger measured along the surface due to shell curvature—this, however, in most specimens is obscured by deformation or breakage). The fourth vertebral contacted the third cranially, the remaining part of the dorsomedial edge of the third pair of pleurals and the whole dorsomedial edge of the fourth pair of pleurals laterally, and the fifth vertebral scute caudally. The widest point of that scute lied in its cranial part. The fifth vertebral was more semi-dome-shaped than the vertebrals first to fourth. It contacted the preceding vertebral cranially and the caudal edges of the last (fourth) pair of pleurals craniolaterally. Laterally and caudolaterally, it contacted the bases of the caudalmost marginals—usually the 12th, the 13th, and the 14th, although sometimes there was no contact with the 12th and at least in ZPAL V.39/48 the 15th pair of marginals was present (see below). Caudally, in Proterochersis spp. there was a caudal notch (Fig. S3). The variability in the vertebral scutes 2–5 is mostly evident medially.

In two small specimens of Proterochersis porebensis (ZPAL V.39/2, Figs. S4A and S4B; ZPAL V.39/34, Figs. 4C and S4C, see Sulej, Niedźwiedzki & Bronowicz, 2012; Szczygielski & Sulej, 2016) a pronounced medial ridge is present crossing the area of the second, the third, and the fourth (in ZPAL V.39/34; in ZPAL V.39/2 this part is missing) vertebral scute. The ridge is rounded to triangular in cross-section, laterally symmetrical, and for most of its length surrounded laterally by deep troughs. Cranially, the ridge and the troughs gradually even out, they tend to nearly disappear in the caudalmost parts of the vertebral scute areas, just before the intervertebral sulci, and in ZPAL V.39/34 the ridge disappears caudally before the throughs do, resulting in a shallow, longitudinal, midline depression in the caudal part of the fourth vertebral scute area (Fig. S4C). The external morphology and small distance between the ribs in ZPAL V.39/2 indicate that it was similar in size to ZPAL V.39/34, which suggests that this morphology of the middorsal keel is related to the young ontogenetic age of the individuals. In Proterochersis porebensis specimens ZPAL V.39/1 (Figs. S4G and S4H), ZPAL V.39/4 (see Szczygielski & Sulej, 2018), ZPAL V.39/72 (Fig. S4D), and ZPAL V.39/169 (Figs. S4E and S4F), and on a small midcarapacial fragment of Proterochersis robusta specimen SMNS 11396, a much more subtle, low, and rounded midline ridge can be seen with equally subtle lateral troughs or no troughs at all. If the middorsal keel of ZPAL V.39/2 and ZPAL V.39/34 is interpreted as a juvenile character, then it seems probable that the middorsal ridges of larger SMNS 11396, ZPAL V.39/1, ZPAL V.39/4, and ZPAL V.39/72 may represent remnants of that structure. No midline ridges can be found in ZPAL V.39/48 (slightly smaller than ZPAL V.39/72 and, judging by rib spacing, comparable in size to ZPAL V.39/1) or ZPAL V.39/49, but the carapaces of these two specimens are broken along the midline, possibly obscuring the ridges. The ridge in ZPAL V.39/1 is slightly tilted cranially to the left, so in the cranial part of the specimen it loses strict correlation with underlying neural processes of the vertebrae (Figs. S4G and S4H). This supports the view that middorsal ridges of proterochersids are not strictly induced by the position of the axial skeleton, but rather are related to epidermal scutes.

Three Proterochersis robusta specimens (CSMM uncat., Fig. 2A; SMNS 17561, Fig. 2F; SMNS 17930, Figs. 2J and 8) exhibit various degrees of indentation along the midline of the second, the third, and the fourth vertebral scute areas. The most severe case is exhibited by CSMM uncat. (Fig. 2A). Along the midline in the cranial regions of the second and the third vertebral, deep, funnel-shaped grooves are present, as if the scute area was cranially split in two. These grooves are approximately as deep as the cranial vertebral sulci with which they are connected, they penetrate the vertebral fields no further than to the mid-length, and caudally they become noticeably shallower and narrower, ending in a sharpened point. In the caudal parts of the scute areas they continue as subtle depressions. The third vertebral lacks the deep groove, but there is a similarly shaped, shallow depression. The fifth vertebral is depressed as well, but the depression is wider and gentle. In SMNS 17930 (Figs. 2I and 6) the anatomy is similar, but less pronounced—there are weak grooves in the cranial parts of the second and the third vertebral area, similar to the caudal sections of the grooves of CSMM uncat., and there is a gentle depression running along the middle of the shell. SMNS 17561 (Fig. 2F) exhibits only a weak depression along the midline, only slightly more pronounced in the cranial sections of the vertebral areas. This morphology initially resembles the midline troughs of ZPAL V.39/2 (Fig. S4A) and ZPAL V.39/34 (Fig. 4A; Fig. S4C) but there are several important differences. Firstly, in CSMM uncat., SMNS 17561, and SMNS 17930 there is no middorsal keel. Secondly, these specimens are relatively large (SMNS 17561 is approx. 35 cm long, SMNS 17930 is approx. 36 cm long, and CSMM uncat. is over 36.5 cm long; note that the damage suffered by SMNS 17930 and CSMM uncat. may cause some underestimation of their sizes and/or relative size differences). Thirdly, the middorsal keels and troughs of ZPAL V.39/2 (Fig. S4A) and ZPAL V.39/34 (Fig. 4A; Fig. S4C) do not reach the cranial edge of the second vertebral and span along the full length of the third vertebral, but do not connect to intervertebral sulci, while the midline grooves or depressions of CSMM uncat. (Fig. 2A), SMNS 17561 (Fig. 2F), and SMNS 17930 (Figs. 2I and 6A–6C) are most pronounced near the cranial edges of the vertebral scutes and in CSMM uncat. they connect to intervertebral sulci. Considering that the vertebral scutes grew mostly in their cranial part (see below), it seems likely that these depressions and grooves developed relatively late during ontogeny, and may be evidence of scute splitting. Congruent with this hypothesis is the observed positive correspondence between the severity of observed morphologies and the size of the specimens. The presence of that morphology on the vertebral scute areas of Proterochersis robusta specimen SMNS 16442 (Fig. 2C) is ambiguous. A shallow groove seems to be present medially, but this specimen is compacted, broken, and its surface is poorly preserved, making assessment difficult.

Figure 6 Proterochersis robusta, SMNS 17930, carapace in (A) lateral right, (B) laterodorsoanterior, and (C) dorsoanterior view.

Note pronounced growth marks.

In Proterochersis robusta specimen SMNS 17930 (Figs. 2I and 6) and in several specimens of Proterochersis porebensis (ZPAL V.39/4, see Szczygielski & Sulej, 2018; ZPAL V.39/34, Fig. 4A; Fig. S4C; ZPAL V.39/49, Fig. 4E; ZPAL V.39/72, Fig. 4G; Fig. S4D; ZPAL V.39/169, Fig. S4E) the sulci separating the first and the second, the second and the third, and/or the third and the fourth vertebral scute area form in the middle small, arrow-shaped cranial projection. In some cases (ZPAL V.39/4, ZPAL V.39/34, ZPAL V.39/72) this projection is laterally surrounded by two rounded caudal projections, resulting in a ω-shaped pattern. The presence of the cranial projection seems to be correlated with, but not exclusive to, the presence of a middorsal keel or ridge.

The intervertebral sulci of most Proterochersis spp. specimens, although relatively straight compared to, for example, circumpleural sulci, exhibit some minor irregularities. In many cases, it is difficult to evaluate whether these irregularities are an effect of post-mortem distortion. Curiously, however, the sulcus between the third and the fourth vertebral seems to be comparatively more prone to severe irregularities. In Proterochersis robusta specimen CSMM uncat. (Fig. 2A) it is clearly asymmetrical in the middle section, where it spans cranially, and an asymmetry of the same sulcus is also profound in another Proterochersis robusta specimen, SMNS 17561 (Fig. 3F)—in that case the sulcus is wavy rather than straight and skewed, so it meets the third pleural more cranially on the right side than on the left. Similarly to CSMM uncat., the mid-section of this sulcus is asymmetrical in Proterochersis porebensis specimen ZPAL V.39/34 (Fig. 4A).

Pleural scutes

Proterochersis spp. had paired rows of four polygonal, slightly elongated pleural scutes (Figs. 1, 2, 4, 6; Fig. S5). The first pleural was hexagonal. In all the specimens of Proterochersis spp. it contacted the first vertebral mediocranially via dedicated facet, and the relative length of this facet seems to increase with the size of the animal (Figs. 2 and 4). In this respect, Proterochersis spp. differed from Keuperotesta limendorsa Szczygielski & Sulej, 2016, in which the sulcus between the first vertebral and the first pleural lies in the same line as the sulcus between the first vertebral and the second marginal, and nearly in the same line as the sulcus between the second vertebral and the first pleural. K. limendorsa, however, is currently represented by a single specimen, so it is difficult to estimate if this difference is taxonomic, ontogenetic, or an effect of intraspecific variability. Beside the first vertebral, the first pleural contacted the second vertebral dorsomedially, the second pleural caudally, the caudal part of the base of the second marginal (with the exception of Proterochersis porebensis specimen ZPAL V.39/57, Figs. S1N and S3B), the whole base of the third, and the cranial part of the base of the fourth marginal as well as the cranial part of the dorsomedial edge of the first supramarginal ventrolaterally, and the second pleural scute caudally. The second pleural was heptagonal and had contacts with the first pleural (cranially), all three supramarginals (ventrolaterally), the third pleural (caudally), and the second and the third vertebral scute (dorsomedially). The third pleural usually was hexagonal, contacted the second pleural (cranially), the third supramarginal scute and the ninth and 10th marginal (ventrolaterally), the fourth pleural (caudally), and the third and fourth vertebral scute (dorsomedially). In most individuals the sulcus with the ninth and 10th marginal was roughly continuous and straight, but in Proterochersis porebensis specimen ZPAL V.39/49 (Figs. 4E and 4F) the basal edges of these scutes were directed at an angle, resulting in a heptagonal third pleural. Less pronounced, but similar condition can be seen also in Proterochersis robusta specimen SMNS 17561 (Fig. 2G) and Proterochersis porebensis ZPAL V.39/72 (Figs. 4G and 4H). The fourth pleural was hexagonal and contacted the third pleural (cranially), the bases of the 10th, 11th, and 12th marginal (in some specimens, such as Proterochersis robusta SMNS 17561, Figs. 2F, 2G; Fig. S5B, and Proterochersis porebensis ZPAL V.39/48, Figs. 4C, 4D; Fig. S5H, the caudalmost tip of the last pleural may also touch the cranial tip of the 13th marginal), and the fourth (dorsomedially) and fifth (caudally) vertebral scute. Usually, the interpleural sulci lack pronounced curvature, but in some specimens (e.g., Proterochersis robusta SMNS 17561, Figs. 2F and 2G, Proterochersis porebensis ZPAL V.39/48, Figs. 4C and 4D, and ZPAL V.39/49, Figs. 4E and 4F) the caudal edges of pleurals 1–3 are slightly concave and form caudal processes in their dorsomedial parts, at the level of pleural tubercles.

Supramarginal scutes

On each side of the carapace there were three elongated supramarginal scutes (Figs. 1, 2, 4 and 6A). The first supramarginal was pentagonal and had its longest tip directed cranially. Caudally, this scute contacted the second supramarginal, and its dorsomedial tip was tucked between the first and the second pleural scute. The second supramarginal was rectangular and contacted the first supramarginal (cranially), the third supramarginal (caudally), and the ventrolateral edge of the second pleural (dorsomedially). The third supramarginal was pentagonal and shaped roughly the same as the first, but with its longest tip directed caudally. This scute contacted the second supramarginal cranially and its dorsomedial tip was tucked between the third and the fourth pleural scute. The row of three supramarginals always contacted the bases of the fifth to ninth marginal ventrolaterally. The intersupramarginal sulci are located roughly at the same level as the sulci separating the sixth, seventh, and eight marginal scute areas, but some several millimeter misalignment frequently occurs—the intermarginal sulci usually are shifted slightly cranially in relation to the intersupramarginal sulci (Proterochersis robusta specimens SMNS 17561, Fig. 2G, SMNS 17755, and SMNS 18440, Fig. 2K; Proterochersis porebensis specimens ZPAL V.39/8 (see Szczygielski & Sulej, 2018); ZPAL V.39/48, Figs. 4C and 4D, ZPAL V.39/49, Figs. 4E and 4F, right side of ZPAL V.39/72, Fig. 4G, ZPAL V.39/160, and, possibly, in ZPAL V.39/34, although the shell margin of that individual is damaged in that region) but in some cases they may be shifted slightly caudally (left side of ZPAL V.39/72, Figs. 4G and 4H, ZPAL V.39/168). Proterochersis porebensis specimen ZPAL V.39/48 is closest to have these sulci coinciding with only few millimeter cranial shift of intermarginal sulci (Figs. 4C and 4D). Other than that, no meaningful variability or clear allometry was observed in the supramarginals. They seem to increase their sizes more or less linearly with the carapace. The largest found supramarginal is the first supramarginal of Proterochersis porebensis specimen ZPAL V.39/8 (see Szczygielski & Sulej, 2018), which was eight cm long and 5.5 cm high—slightly larger than in ZPAL V.39/49 (7.7 × 5 cm) and significantly larger than in ZPAL V.39/48 (6.6 × 4.2 cm) and ZPAL V.39/72 (6.7 × 4.2 cm). Unfortunately, the first supramarginal is too severely damaged in ZPAL V.39/34 to allow precise measurement, but the probable outline of this scute on the right side of the specimen suggests the length of approx. 4.2 cm. This would mean that, even more so than for the ninth marginal, there is a good correlation between the length of the shell and the length of the first supramarginal (Pearson correlation coefficient = 0.994 for n = 4). Based on that, the shell of ZPAL V.39/8 may be estimated to be over 50 cm long.

Marginal scutes

There were two rows of marginals spanning from the craniolateral region of the cervical scute to the caudolateral limits of the caudal notch (Figs. 1–5; Fig. S5). Typically, each row included fourteen scutes. Besides some minor random variations in shape and relative size, which are usually difficult to grasp, the marginal scutes of Proterochersis spp. exhibited variability in three main ways.

Firstly, their number was variable—variants of 15 marginals (ZPAL V.39/48—possibly a supernumerary abnormality) and 14 marginals (all the other specimens with complete marginal series) are known (see Szczygielski & Sulej, 2016). There are at least 12 marginals identifiable in the juvenile ZPAL V.39/34, but their exact number is uncertain due to preservation, so it is probable that the typical number of 14 marginals was present. The layout relative to pleurals and supramarginals suggests that one intermarginal sulcus is likely to be undetected in the bridge region, below the supramarginal row, and this area is heavily damaged on both sides of ZPAL V.39/34. Another likely missing sulcus should be located craniolateral to the cervical scute and should delineate the first marginal. This area, however, is well-preserved in ZPAL V.39/34. It is, nonetheless, possible that the scute was there, but its sulcus is too subtle to be identified (many sulci on that specimen are very weak, see below) or that in such a young animal the scute was very small and located at the very edge of the carapace—possibly the first marginal exhibited allometry during development. This option seems plausible mainly because there is no nuchal notch in ZPAL V.39/34 (the cranial edge of the carapace is flush—Figs. 4A and 4B, see also Sulej, Niedźwiedzki & Bronowicz, 2012; Szczygielski & Sulej, 2016) in some specimens (particularly SMNS 17561, Fig. 2F, ZPAL V.39/48, Fig. 4C, ZPAL V.39/49, Fig. 4E, and on the right side of ZPAL V.39/72, Fig. 4G) the first marginal scute was almost entirely cranial to the cervical scute (and, optionally, to the second marginal), having very little or no contact with the first vertebral scute. This leaves two possibilities—either the first marginal scute was in some individuals “crowded out” during ontogeny by the cervical and the second marginal or, at least in some individuals, it started to develop on the very margin of the shell. Alternatively, some variability in the number of marginal scutes is possible. Note that this condition is different from the missing first marginal of K. limendorsa—in K. limendorsa the lateral contact between the cervical scute and the marginal series is very narrow or nonexistent (Szczygielski & Sulej, 2016), while in ZPAL V.39/34 it is wide. The smallest fragmentary specimen with the first marginal craniolateral to the cervical scute is ZPAL V.39/390 (Figs. S1L and S1M).

The second type of marginal variability of Proterochersis spp. affects the layout of the intermarginal sulci relative to the sulci of the remaining scutes, resulting in (usually minor) differences of contacts between these scutes and variation of their shape. The first marginal in dorsal aspect always contacted the cervical caudomedially, was subtriangular or trapezoidal, depending on whether it was prevented from the contact with the first vertebral by the lateral tips of the cervical scute (as in Proterochersis porebensis ZPAL V.39/48, Fig. 4C, ZPAL V.39/49, Fig. 4E, ZPAL V.39/57, Figs. S1N and S3B, and ZPAL V.39/72, Fig. 4G) or not (as in Proterochersis robusta SMNS 17561, Fig. 2F, and SMNS 17930, Figs. 2I, 6B and 6C, and Proterochersis porebensis ZPAL V.39/22, Fig. S3A), respectively, and had a rounded craniomedial tip. In ventral aspect, it was subtriangular and had a concave base. In this aspect, the intermarginal sulci of this and the following nine marginals as well as the basal sulci of all except the first marginal are gently convex. The second marginal was subrectangular to trapezoidal both in dorsal and in ventral aspect, always contacted the first vertebral, in some specimens its tip touched the cervical (Proterochersis porebensis ZPAL V.39/48, ZPAL V.39/49, ZPAL V.39/57, and ZPAL V.39/72), and in ZPAL V.39/57 (Figs. S1N and S2B) it also touched the first pleural. Consequently, in most Proterochersis spp. specimens the third marginal scute was pentagonal in dorsal aspect (subrectangular in ventral aspect) and contacted both the first pleural and the first vertebral scute (Figs. 2 and 4), but ZPAL V.39/57 (Figs. S1N and S3B) is the only known exception—the sulcus between the second and the third marginal scute in that specimen is continuous with the sulcus between the first vertebral and the first pleural scute, and the third marginal was subrectangular in dorsal aspect. The fourth marginal was always subrectangular in both aspects and contacted the first pleural in dorsal aspect and cranial part of the axillary in ventral aspect. The fifth marginal was always pentagonal both in dorsal ventral aspect, and contacted both the first pleural and the first supramarginal dorsomedially, and the axillary and the first inframarginal scute ventromedially. In dorsal aspect, the marginals sixth to eighth always contacted the row of three supramarginals and were subrectangular to weakly pentagonal (depending on how much their intermarginal sulci are offset from the intersupramarginal sulci, see above). In ventral aspect, they are usually pentagonal and contact the row of four inframarginals. The caudal sulcus of the sixth marginal in this aspect is located around the level of the sulcus between the first and the second inframarginal—in Proterochersis robusta specimen SMNS 17561 (Fig. 3F) and Proterochersis porebensis specimen ZPAL V.39/48 (Fig. 5B) it is slightly caudal, but seems to be slightly cranial in ZPAL V.39/49 (although the exact morphology is obscured in that individual by damage, Fig. 5C), and it falls on a gap between the inframarginals in Proterochersis robusta SMNS 18440 (Fig. 3H) and in Proterochersis porebensis ZPAL V.39/21 (see below). The caudal sulci of the seventh and the eighth marginal fall around the midlengths of the third and the fourth inframarginal, respectively. The ninth marginal was pentagonal in dorsal aspect and subpentagonal in ventral aspect, gradually increasing its size caudally. It contacted the third supramarginal dorsomediocranially, and the third pleural dorsomedially. The ventromedial edge was gently curved rather than angular, it contacted the fourth inframarginal and formed the caudal end of the bridge. The 10th marginal was pentagonal in dorsal aspect and subrectangular in ventral aspect. Dorsomedially, it contacted the third and the fourth pleural. In most cases, the dorsomedial sulcus of the 10th marginal is roughly continuous with the dorsomedial sulcus of the ninth marginal, although in Proterochersis porebensis specimen ZPAL V.39/49 (Figs. 4E and 4F) these sulci are set at an angle. A similar, but less pronounced break in sulcus direction is also present in Proterochersis robusta specimen SMNS 17561 (Fig. 2G) and Proterochersis porebensis ZPAL V.39/72 (Figs. 4G and 4H). The 11th marginal was always subrectangular both in dorsal and in ventral aspect and dorsomedially it contacted the fourth pleural. The 12th marginal was either trapezoidal (dorsomedial contact with fourth pleural only—Proterochersis robusta SMNS 17561, Figs. 2F and 2G; Fig. S5B, Proterochersis porebensis ZPAL V.39/48, Figs. 4C and 4D and S5H) or pentagonal (dorsomedial contact with the fourth pleural and the fifth vertebral—remaining specimens, Figs. 2 and 4; Fig. S5) in dorsal aspect due to the varied position of the sulcus between the 12th and the 13th marginal relative to the sulcus between the fourth pleural and the fifth vertebral scute area (see Szczygielski & Sulej, 2016). In SMNS 17561 (Figs. 2F and 2G; Fig. S5B), ZPAL V.39/48 (Figs. 4C and 4D; Fig. S5H), ZPAL V.39/72 (Figs. 4G and 4H; Fig. S5K), and ZPAL V.39/386 (Fig. S5N) these sulci are located nearly in the same line (the intermarginal sulcus usually only slightly caudal, but on the left side of SMNS 17651 even slightly cranial), while in SMNS 17755a (Fig. 2H; Fig. S5C) and ZPAL V.39/49 (Fig. S5I) the pleurovertebral sulcus falls close to the middle of the 12th marginal, and the intermarginal sulcus is located clearly more caudally. This is also the configuration of sulci in the corresponding region of ZPAL V.39/34 (Figs. 4A and 4B; Fig. S5G), regardless of the number of marginals in that specimen. In SMNS 17930 (Figs. 2I, 2J and 6A), the sulcus between the last pleural and the last vertebral lies approximately in the same line as the intermarginal sulcus between the 12th and the 13th marginal but the pleurovertebral sulcus in that specimen is fully contained in the area restored with plaster and has an unusual layout (it is continuous with the sulcus between the last pleural and the fourth vertebral instead of creating an angle, as in other specimens—compare Figs. 2 and 4), so it seems more plausible that in life it met the 12th marginal in the middle. Given the limited sample which still exhibits some variance in the relative position of sulci, it is possible that these two morphologies are not the only possibilities, but a full spectrum of intermediate morphologies existed in the population. Regardless of the shape of the 12th marginal, the 13th marginal was always subtrapezoidal in dorsal aspect, had a convex rim, and contacted the fifth vertebral dorsomedially (in SMNS 17561, Figs. 2F and 2G; Fig. S5B) and ZPAL V.39/48 (Figs. 4C and 4D; Fig. S5H) additionally touching the caudal end of the fourth pleural). Both the 12th and the 13th were subrectangular in ventral aspect. In most specimens, the 14th marginal is the last of the series and in subadult and adult specimens it had a rounded or spiky rim, the end of which was free from the preceding marginal. In ZPAL V.39/48 this morphology is exhibited by the 15th marginal, while 14th is intermediate between the 15th and the 13th (Figs. 4C and 4D; Fig. S5H). In some specimens (ZPAL V.39/6, Fig. S3D, ZPAL V.39/18, Fig. S5E, ZPAL V.39/48, Fig. S5H) the sulcus between the last and second-to-last marginal is sinuous. The notch between the last two marginals in most specimens (except ZPAL V.39/23, Fig. S5D, ZPAL V.39/72, Fig. S5K, and ZPAL V.39/380, Fig. S5M) is rounded and the bone around the level of the sulcus or just caudal to it is thinner than in the middle of the marginal areas. Dorsomedially, the 14th and the 15th marginal (if present) contacted only the fifth vertebral.

The caudalmost marginals (be it the 14th or the 15th) grew in a characteristic manner. In Proterochersis porebensis specimen ZPAL V.39/34 (Fig. S5G) and in Proterochersis robusta specimen SMNS 17561 (Fig. S5B) the last pair of marginals was small and triangular (they lacked a caudoventral tip on their rims), broader then long (in ZPAL V.39/34 2.1 cm wide, measured along the sulcus with the last vertebral and one cm long in the longest place; not measured in SMNS 17561), and their edge was continuous with the edge of the preceding pair, resulting in lack of serration (see also Szczygielski & Sulej, 2016). Proterochersis porebensis ZPAL V.39/23 (Fig. S5F) is the smallest last marginal that has a tip, resulting in its roughly rhomboidal shape. It is 1.6 cm wide, its maximal size (measured from the tip to the corner of the sulci with the fifth vertebral and preceding marginal) is 1.9 cm, and length (from the sulcus with the fifth vertebral to the tip, parallel to the caudal edge) is 1.4 cm (although the tip is broken, so these measurements should probably be about one mm larger). Slightly larger (last marginal 2.1 cm wide, two cm long, 2.4 cm max. size) Proterochersis porebensis individual, ZPAL V.39/18 (Fig. S5E), exhibits a transitional morphology linking these small specimens and the more adult-like morphology—there is a small but distinct tip and a shallow but noticeable rounded notch separates it from the rim of the preceding marginal. In larger (and, supposedly, older) individuals, the last marginals were becoming spikier, and longer than wide. The largest last marginal found thus far is in Proterochersis porebensis specimen ZPAL V.39/59 (Fig. S5J; its width is 3.2 cm, maximal size is 5.1 cm, and length is 3.9 cm). The remaining caudal marginals in large specimens, as evidenced by ZPAL V.39/6 (Fig. S5D) and ZPAL V.39/59 (Fig. S5J), also tended to increase their sizes toward the periphery of the shell, but lacked the serration and spikiness of the last marginal.

One of the largest fragmentary Proterochersis porebensis specimens, ZPAL V.39/60 (Figs. S1O and S1P), has the ninth marginal 7.8 cm long (measured on the external side of carapace, close to the edge). ZPAL V.39/34 has this marginal approximately 3.5 cm long, ZPAL V.39/48—5.2 cm long, ZPAL V.39/49—6.5 cm long, and ZPAL V.39/72—6 cm long. There seems to be reasonably good correlation between the length of carapace (see above) and the length of that element (Pearson correlation coefficient = 0.986 for n = 4). Based on that, the shell of ZPAL V.39/60 may be estimated to reach up to about 60 cm in length.

Scute sulci and surface

The morphology and size of sulci in carapaces of Proterochersis spp. is dependent on their ontogenetic age, as inferred from shell size and morphology. There is a positive correlation between the size of the animal and the depth and breadth of sulci. In ZPAL V.39/34 (Figs. 4A, 4B; Figs. S4C, S5G) the sulci on the carapace are less than 1 mm wide and in some cases one edge of the sulcus (e.g., the caudal edge of the vertebral scute area in intervertebral sulci) is slightly curved externally, creating a characteristic lip and making it a bit higher than the other edge (rarer is the situation when both the edges are raised, as in ZPAL V.39/2, Fig. S4A). Also in ZPAL V.39/34, some sulci (e.g., between some lateral marginals) are very poorly defined or seem to be bulging rather than sunken (e.g., between the cervical and the cranial marginals or between the supramarginals)—the latter morphology may be a combination of the two former, that is, the sulcus proper (the groove) is too weak to be seen, but the lip around the periphery of one of the scutes is visible. In larger specimens the sulci are broader (over one centimeter in ZPAL V.39/63, Fig. S1B) and always sunken. The intervertebral and intermarginal sulci usually have their cranial edge (formed by the preceding scute area) slightly higher than the caudal one (formed by the succeeding scute area), but the edges are usually rounded and rarely form a curved lip (e.g., ZPAL V.39/169, Fig. S4E; see also peripherals and costals figured in Szczygielski & Sulej, 2018).

Most scute sulci on the carapace of Proterochersis spp. are sinuous (undulating; Figs. S8A, S8B and S8E–S8G). This, however, seems to be at least partially determined by the size of the individual—in probable juveniles, such as SMNS 16603 (Figs. 2D–2E) and ZPAL V.39/34 (Figs. 4A, 4B; Figs. S4C and S5G), the sulci appear to be straight, and with age their undulation increases. It is most prominent around the supramarginals and pleurals (Figs. S8A and S8B). The undulation is also related to the radial striation on the surface of the scutes (Figs. S8A–S8B, S8E and S8F), which is frequently visible (although usually faint) as imprints on the bone surface. The surficial striation and the undulation of sulci are most prominent in the carapaces of the largest specimens, such as SMNS 16442, ZPAL V.39/49 (Figs. 4E and 4F), ZPAL V.39/59 (Fig. S5J), and ZPAL V.39/63 (Fig. S1B), because the frequency of the undulations seems to decrease and the depth of the striations as well as the amplitude of the undulations seem to increase with growth (Fig. S8G). The striation on the pleurals is most prominent along their cranial and pleuromarginal sulci (Figs. S8A and S8B), where the grooves are longer than along the caudal and pleurovertebral sulci. Most marginals of not very large individuals exhibit weak undulation of sulci and striation, with the exception of the ninth marginal, in which these characters are strongly expressed along the sulcus with the third supramarginal. Usually, the intervertebral sulci do not undulate (even though the pleurovertebral sulci and the craniolateral sulcus of the first vertebral scute are clearly sinuous and, especially the latter, frequently exhibit striation), but in very large individuals (e.g., ZPAL V.39/63, Fig. S1B) the intervertebral sulci are becoming slightly uneven. Separate from the radial striation are the bowed, concentric growth marks (Figs. S8A–S8F). These marks are located in the same areas as the radial striation (most notably on pleurals along the cranial pleural and pleuromarginal sulci and on vertebrals along the pleurovertebral sulci and along the craniolateral sulcus of the first pleural), but are parallel rather than perpendicular to the scute sulci, usually fainter, broader, and less densely packed. They do not reach the borders of the scute, and thus are not correlated with the undulation of the sulci. Their relatively large breadth and shallowness makes them difficult to spot on supramarginals and on the caudal and dorsomedial parts of the pleurals. Together with their low number even in large specimens (no more than five pronounced growth marks per scute are observed in available material) and their absence in young specimens, this also indicates that they do not exhibit strict seasonal (annual) iterativity, but rather developed as a result of long-term (polyseasonal) changes of environmental conditions. Both the radial striations and the growth marks seem to originate near the caudodorsomedial region of the pleurals, where the bone is thickened to a boss. This agrees with the observed pattern of scute growth (see below). A similar boss is also present in some specimens in the caudodorsomedial region of the first supramarginal, near the dorsomedial edge of the second supramarginal, and in the craniodorsomedial region of the third supramarginal.

Proterochersis robusta specimen SMNS 17930 (Figs. 2I and 6; Fig. S8D) is unique in its accentuated growth marks of its vertebral and pleural scutes. There are two generations of these abnormal growth marks per scute and they are bilaterally symmetrical. In breadth and position they resemble typical growth marks of other Proterochersis spp. specimens (such typical growth marks are also present between and above the abnormal ones in SMNS 17930, Fig. 6) but they are deeper (in that respect approximating sulci) and have sharper edges. Along the cranial edges of the first and the third vertebral scute the growth marks of the older, higher positioned generation are bilaterally continuous and take form of “fake sulci” by copying the shape of true sulci in front of them (albeit in smaller scale, as evidenced by the first vertebral). They, however, do not reach the edges of the scutes and do not connect to true sulci. Based on the fact that this morphology is present only in this one, middle-sized specimen, we interpret it as pathological.

Plastron

Small specimens

Several fragmentary specimens of small plastral bones morphologically resembling early stages of plastra development in modern turtles are known from the Proterochersis spp.-yielding localities of Murrhardt and Poręba—SMNS 81917 (Fig. S6A), ZPAL V.39/165, ZPAL V.39/197 (Fig. S6C), ZPAL V.39/277 (Fig. S6B), ZPAL V.39/383, ZPAL V.39/384 (Fig. S6D), and several other specimens from Poręba. Given their size (less than eight cm each), it is likely that they belong to young juveniles, older than hatchlings but younger than ZPAL V.39/34, which has its shell completely ossified. Other than the typical characteristics of developing plastral bones—jagged edges with fingerlike projections and minute striation indicative of progressing intramembranous ossification (e.g., Gilbert et al., 2001)—they exhibit few superficial characters, no identifiable sulci, and only ZPAL V.39/165 and ZPAL V.39/197 (Fig. S6C) can be identified with relative confidence as hyoplastra, based on the shape of their incipient axillary buttresses. SMNS 81917 (Fig. S6A) is up to two mm thick and has a rounded notch, which indicates that it is either a hyoplastron or a hypoplastron. Unfortunately, it is exposed only in visceral view and flattened, therefore it is difficult to determine the anatomical orientation of that specimen with full confidence. For that reason, it is also difficult to identify it more precisely. A lip along one of the edges of the notch and gentle thickening along the other edge differentiate this specimen from ZPAL V.39/165 and ZPAL V.39/197, potentially hinting that it is a hypoplastron. Based on the overall shape and relatively large thickness (five to eight mm, compared to one to maximally five mm of ZPAL V.39/165 and ZPAL V.39/197), ZPAL V.39/277 is likely a xiphiplastron (compare to, Zangerl, 1939; Gilbert et al., 2001; Rice et al., 2016) or may be one of the mesoplastra—it is thicker than ZPAL V.39/165 and ZPAL V.39/197, even though they are larger and relatively well-developed, so it is unlikely that this element represents a hyoplastron at an earlier developmental stage, and for the same reason its identity as a hypoplastron may be likely refuted. Overall, the developing plastral bones which may be attributed to Proterochersis spp. are already more similar to plastral bones of derived turtles than to fusing gastralia, from which the plastron is thought to have originated (Schoch & Sues, 2015, 2017).

Gular and extragular scutes

Proterochersis spp. had a pair of gular (roughly pentagonal in ventral view) and extragular (roughly trapezoidal in ventral view) scutes located at the cranial end of the plastron (Figs. 1, 3, 5, 7; Fig. S8), contacting the cranial edges of the humeral scutes (the exception being Proterochersis porebensis ZPAL V.39/385, see below). The caudal sulci of the gulars are roughly straight or gently concave and skewed craniolaterally, while the caudal sulci of the extragulars are gently convex and skewed caudolaterally. Usually, the gulars are separated from the extragulars by a tilted, craniolaterally directed sulcus, but the angle of tilting varies between specimens and in SMNS 16603 the sulcus is directed craniocaudally. It appears that the size of gulars relative to extragulars is variable—for example, in ZPAL V.39/48 they are, respectively, 2.6 and 3.2 cm wide, while in ZPAL V.39/385, 1.9 and 3.9 cm wide (measured cranially).

The shape of the gulars and extragulars shows large variability, some (but not all) of which may be explained by the ontogeny. In smallest (and, supposedly, youngest) specimens, such as Proterochersis robusta SMNS 16603 (Fig. 3E; Fig. S7B) and Proterochersis porebensis ZPAL V.39/34 (Fig. 5A; Figs. S7E–S7G) these scutes were nearly completely flat ventrally (their ventral surface was flat and more or less parallel to the humerals behind them) and the cranial edge of the cranial plastral lobe was nearly flush (see Szczygielski & Sulej, 2016). In all specimens larger than SMNS 16603 and ZPAL V.39/34, the gulars and extragulars had cranial or craniolateral tubercle-like projections, due to their length in lateral parts increasing proportionally faster than in the medial parts, resulting in characteristically scalloped cranial edge of the cranial plastral lobe and the gulars typically ending in a rounded point (Figs. 3, 5, 8; Fig. S7). This character seems to be clearly related to the size of the individual. This craniolateral expansion is accompanied by thickening of the gulars (most prominent craniomedially). In Proterochersis robusta specimens that preserve this region (CSMM uncat., Figs. 3A and S7A; SMNS 16442, Fig. 3D; SMNS 17561, Fig. 3F; Figs. S5C and S5D; ZPAL V.39/388, S5K′–S5M′) the gulars and extragulars (in case of SMNS 17561, the only one preserving these scutes) remain relatively flat, only gently bulged ventrally. In most Proterochersis porebensis specimens, however, the gulars are significantly thickened in their craniomedial parts, so most of their ventral surface is skewed caudolaterodorsally (ZPAL V.39/48, Fig. 5B; Figs. S7H–S7J; ZPAL V.39/49, Fig. 5C; Figs. S7K–S7M; ZPAL V.39/187, Figs. S7R–S7T; ZPAL V.39/189, Figs. S7U–S7X; ZPAL V.39/333, Figs. S7Y–S7A′; ZPAL V.39/379, Figs. S7B′–S7D′; ZPAL V.39/387, Figs. S7H′–S7J′; ZPAL V.39/388, Figs. S7K′–S7M′; ZPAL V.39/420, Figs. S7N′–S7P′; see also Articles S2–S3). In ZPAL V.39/187 (Figs. S7R–S7T) the gular was apparently relatively narrow and its point was located around its mid-section rather than closer to the lateral border of the scute, resulting in seemingly lower angle between the gular projections—the specimen is, unfortunately, incomplete, so its exact orientation is uncertain. ZPAL V.39/385 (Fig. 7; Figs. S7E′–S7G′) differs from the remaining specimens in that its gulars are rounded ventrally and cranially. The preserved medial parts of gulars of SMNS 16442 (Fig. 3D) are flat and proportionally thin despite its relatively large size, but this may be caused by the compaction and/or related to the fact that in that specimen the epiplastra, which formed a large section of the gular tubercles, are missing (Szczygielski & Sulej, 2018).

Figure 7 Proterochersis porebensis, ZPAL V.39/385, anterior plastral lobe with supernumerary scutes (*) in (A) visceral and (B) external view.

The extragulars in the specimens larger than SMNS 16603 and ZPAL V.39/34 tend to lose their alignment with the humerals, and instead become inclined more ventrally (i.e., toward the external surface of the plastron; note that the cranial plastral lobe itself was bent more or less dorsally, so in life this ventral inclination of the extragulars relative to the humerals resulted in their free ends pointing predominantly cranially rather than ventrally; see Fig. 1B). The angle between the cranial end of the humeral scute and base of the extragular scute at the level of the extragulohumeral sulcus is variable, in some specimens being low (e.g., ZPAL V.39/49, Figs. 5C; Figs. S7K–S7M; ZPAL V.39/333, Figs. S7Y–S7A′), but much more distinct in others (e.g., ZPAL V.39/385, Fig. 7; Figs. S7E′–S7G′; ZPAL V.39/387, Figs. S7H′–S7J′). In addition to the angle at the base of the extragular, the extragular exhibits various degrees of ventral curvature (the dorsal surface is convex dorsally, the ventral surface concave ventrally). In some specimens (e.g., SMNS 17561, Figs. 3F, S5C and S5D; ZPAL V.39/333, Figs. S7Y–S7A′; ZPAL V.39/385, Figs. 7 and S7E′–S7G′; ZPAL V.39/388, Figs. S7K′ and S7M′) this curvature is minimal or virtually absent and thus, if the angle between the base of the extragular and the humeral was also low, the cranial end of the extragular was only slightly turned ventrally (e.g., ZPAL V.39/333, Figs. S7Y–S7A′; ZPAL V.39/388, Figs. S7K′–S7M′). In other specimens the curvature is much more prominent and as a result the angle between the tip of the extragular and the cranial portion of the humeral approximated 70° (e.g., ZPAL V.39/48, Fig. 5B; Figs. S7H and S7J; ZPAL V.39/49, Fig. 5C; Figs. S7K and S7M; ZPAL V.39/387, Figs. S7H′–S7J′). The downward curve of the extragulars may be easily explained by their growth pattern (the dorsal part of the scute growing faster than the ventral part), hinting that it was acquired with age. There is, however, no correspondence between the size of the individual and the angle between the extragular tip and the humeral—ventrally flat extragulars set at a low angle relative to the humeral are found both in small (e.g., ZPAL V.39/388, Figs. S7K′–S7M′) and large (e.g., ZPAL V.39/187, Figs. S7R–S7T; ZPAL V.39/333, Figs. S7Y–S7A′) individuals. Likewise, large angle between the extragular tip and the humeral is present both in small (ZPAL V.39/387, Figs. S7H′–S7J′) and large (ZPAL V.39/49, Fig. 5C; Figs. S7K–S7M) specimens. This independence from the individual size may point at sexual dimorphism, but The PCA plots and the regression analyses performed by us do not support that interpretation, suggesting that the observed range of morphologies represents an intraspecific variability instead (see below). The extragulars also differ in the shape of their craniolateral edge, which ranges from a pronounced point (most evident in ZPAL V.39/420, but also present in ZPAL V.39/48, Fig. 5B; Figs. S7H–S7J; ZPAL V.39/49, Fig. 5C; Figs. S7K–S7M; ZPAL V.39/186, Figs. S7N–S7Q; and ZPAL V.39/379; Figs. S7B′–S7D′). In ZPAL V.39/187 (Figs. S7R–S7T), ZPAL V.39/333 (Figs. S7Y–S7A′), ZPAL V.39/385 (Figs. 7; Figs. S7E′–G′), and ZPAL V.39/388 (Figs. S7K′–S7M′) the angle between the cranial and the lateral edge of the extragular is closer to 90°. In ZPAL V.39/387 (Figs. S7H′–S7J′) these edges meet at an obtuse angle, like in juveniles.

The gular and extragular tubercles of Proterochersis robusta known to date show less pronounced features (thickening, ventral curvature, prominent craniolateral points of extragulars) than those of Proterochersis porebensis. It remains unclear whether this is an artifact of a small sample (only four specimens of Proterochersis robusta have this region preserved, including juvenile SMNS 16603 and very incomplete CSMM uncat. and SMNS 16442) or if the more exaggerated morphologies in adult specimens are an autapomorphy of Proterochersis porebensis.

In ZPAL V.39/49 (Fig. 5C; Figs. S7K–S7M) the gulars are laterally asymmetrical (the right one is 2.7 cm wide while the left one is 2.4 cm wide) even though the intergular sulcus is located on the midline of the cranial plastral lobe. In CSMM uncat. (Fig. 3A; Fig. S7A) and SMNS 16603 (Fig. 3E; Fig. S7B) this sulcus is slightly moved to the left. Only the right gular is preserved in these two specimens, but it may be hypothesized that this also caused some minor asymmetry.

Abnormal scute set

Proterochersis porebensis specimen ZPAL V.39/385 (Fig. 7; Figs. S7E′–S7G′) exhibits a scute abnormality. In this specimen, between the row of gulars and intergulars and the set of humerals, there were paired, roughly triangular supernumerary scutes. The right one was slightly smaller than the left one, and did not reach the lateral edge of plastron, allowing partial contact between the right extragular and humeral. The left, larger supernumerary scute did reach the edge of the plastron, and thus separates the extragular from the humeral completely. The sulci separating these two additional elements from the humerals have slightly raised edges, are deepest medially, and become less clear laterally. At first sight, their layout resembles the caudolateral suture of the entoplastron, as visible in Proterochersis robusta specimen SMNS 16442 (Fig. 3D; see Szczygielski & Sulej, 2018), but upon closer inspection, they cannot be mistaken for this suture—their edges are smooth, they lack interdigitation and other macro- and microscopic characteristic of sutures but instead their morphology is consistent with that of other sulci in that specimen, they are located more caudolaterally than the entoplastral suture and do not enter the area of gulars (nor extragulars), and there is no sign of sutures on the visceral surface of that specimen.

Humeral scutes

Proterochersis spp. had a set of two humerals (Figs. 1, 3, 5, 7; Fig. S7) located caudal to the gulars and extragulars (except ZPAL V.39/385, see above) cranial to the pectorals. The caudal sulci of the humeral set have a characteristic appearance—their lateral ends are turned cranially, and medial ends are usually turned more or less caudally, forming a variably pronounced tip (best visible in Proterochersis robusta specimens CSMM uncat., Fig. 3A, and SMNS 17561, Fig. 3F, as well as in Proterochersis porebensis specimen ZPAL V.39/49, Fig. 5C). Besides ZPAL V.39/385, which had the craniomedial edges of the humerals misshaped due to presence of an additional abnormal scute pair, there is no clear variability in humeral shape.

Axillary scutes

There was a pair of elongated, hexagonal axillary scutes present in Proterochersis spp. Each contacted the ventromedial bases of the fourth and fifth marginal, the cranial border of the first inframarginal, and the cranial edge of the lateral part of the pectoral scute. Best preserved in Proterochersis robusta SMNS 17561 (Fig. 3F) and Proterochersis porebensis ZPAL V.39/48 (Fig. 5B), these scutes do not exhibit visible variation.

Pectoral scutes

There was a pair of pectoral scutes present in Proterochersis spp. (Figs. 3, 5 and 7). Cranially, they contacted the humerals, their lateral ends contacted with the axillaries and the first two pairs of inframarginals, and caudally they contacted the first pair of abdominal scutes. The only specimen with an unusual shape of the pectorals is Proterochersis robusta specimen SMNS 17561 (Fig. 3F), in which the scutes were abnormally elongated caudally in the middle portion (see below).

Abdominal scutes

There were two pairs of wide and short, strap-like abdominals in Proterochersis spp. (Figs. 1, 3 and 5). The first pair was located between the pectorals (cranially), the second and the third inframarginal (laterally), and the second pair of abdominals (caudally). The latter, beside the first abdominal pair, contacted the third and fourth inframarginal and the ninth marginal laterally, and the femoral scutes caudally. Both abdominal pairs gradually increased in length toward the ventromedial end of the bridge, at which level their cranial sulci are characteristically bent. From that point toward the lateral ends of the bridge, the length of the first abdominal remained roughly constant and the length of the second abdominal slightly decreased. Typically, abdominals of both pairs met medially. In Proterochersis robusta specimen SMNS 17561 (Fig. 3F), however, the first pair of abdominals lacked medial contact. These scutes gradually decreased in length toward the midline and disappeared completely just before reaching it. Their place in the mid-section of the plastron seems to be partially taken by the caudal expansions of the pectorals and partially by the second pair of abdominals, which seem to retain roughly constant length instead of tapering medially, as typical (compare Figs. 3, 5 and 7). This morphology is known in only one individual out of at least five other specimens of Proterochersis robusta and several specimens of Proterochersis porebensis. We consider it as an abnormality, although the small sample size and retained symmetry make this assumption tentative. SMNS 15479 (a double external mold of proterochersid plastron found in Reichenbach, Germany, figured but not described by Gaffney, (1990, fig. 68) therein) lacks characters that would allow its precise identification as Proterochersis robusta, but such an identity is possible, in which case it would further support the medial contact of the first pair of abdominals as the norm.

Inframarginal scutes

Proterochersis spp. had four polygonal or rounded inframarginal scutes on each side (Figs. 1, 3 and 5). Dorsolaterally, they contacted the marginal row (fourth to eighth marginal, see above). Cranially, the first inframarginal contacted the axillary scute. Ventromedially, the inframarginals contacted the lateral ends of the pectoral (first and second inframarginal), the first abdominal (second and third inframarginal), and the second abdominal (third and fourth inframarginal) scute. In ZPAL V.39/34 (Fig. 5A), the inframarginals were relatively narrow, elongated, and comma-shaped. In larger specimens (SMNS 17561, Fig. 3F; SMNS 17755, Fig. 3G; SMNS 18440, Fig. 3H; ZPAL V.39/48, Fig. 5B; ZPAL V.39/49, Fig. 5C) they increasingly gained girth, becoming relatively flatter. In Proterochersis robusta specimen SMNS 17755 (Fig. 3G) the third and the fourth inframarginal are separated by a small gap. In Proterochersis robusta specimen SMNS 18440 (Fig. 3H) there is a triangular gap between the cranial part of the third inframarginal and the seventh marginal, and possibly there was a gap between the first inframarginal, the sixth and the seventh marginal, and (maybe) the second inframarginal, but the caudal part of the first marginal is damaged, making this uncertain. In Proterochersis porebensis ZPAL V.39/21 there also is an apparent gap between the first and the second inframarginal, around the level of the sulcus between the fifth and the sixth marginal. These gaps seem to lack any pores inside, so they likely were interplates covered by skin rather than housed Rathke’s glands, especially since there is no evidence of similar gaps in the remaining specimens of Proterochersis spp.

Femoral scutes

The femorals in Proterochersis spp. were located caudal to the bridge, between the second pair of the abdominal scutes and the anals (Figs. 1, 3 and 5). Typically, the abdominofemoral sulcus is gently bowed caudally, and in most specimens (the exceptions being Proterochersis robusta SMNS 16442, Fig. 3C, and Proterochersis porebensis ZPAL V.39/48, Fig. 5B) this is also true for the femoroanal sulcus. Both of these sulci are always directed caudolaterally—the femoroanal sulcus more profoundly than the abdominofemoral. No clear variability is observed in these scutes.

Figure 8 Results of the Principal Component Analysis of the gular plastral region in (A and B) ventral view and (C–F) vertical cross-section of extragulars.

The main shape changes for each PC are represented by wireframes, black wireframe is the average shape, while the red line represent shape changes connected to adequate PCs. Proterochersis robusta is represented by stars, P. porebensis by dots. Representations of scute shapes not to scale.

Figure 9 Variability of caudal region of plastron of (A–C) Proterochersis robusta and (D–T) P. porebensis.

(A) CSMM uncat.; (B) SMNS 12777; (C) SMNS 17561; (D) ZPAL V.39/49; (E) ZPAL V.39/71; (F) ZPAL V.39/69; (G) ZPAL V.39/48; (H) ZPAL V.39/68; (I) ZPAL V.39/70; (J) ZPAL V.39/66; (K) ZPAL V.39/34; (L–N) ZPAL V.39/56, left caudal process in (L) dorsal, (M) ventral, and (N) lateral view; (O–Q) ZPAL V.39/199, left caudal process in (O) dorsal, (P) ventral, and (Q) lateral view; (R–T) ZPAL V.39/200, (?)left caudal process in (R) dorsal, (S) ventral, and (T) lateral view. (A–K) in ventral view, in the same scale, ordered roughly by decreasing size. (L–T) in the same scale.

Anal scutes

Contacting the femorals cranially and the intercaudal and caudal scutes caudally, the anals of Proterochersis spp. were the longest scutes in the caudal plastral lobe (Figs. 1, 3 and 5). They gradually decreased in width caudally. No clear variability was observed for these scutes.

Intercaudal and caudal scutes

The caudalmost part of the plastron, presenting a set of two caudal and one intercaudal scutes, seems to be the most variable section of the shell in Proterochersis spp. (Figs. 1, 3, 5 and 9). In the smallest specimens, such as ZPAL V.39/34 (Figs. 5A and 9K) and ZPAL V.39/66 (Fig. 9J), the caudal processes are small, wider than long, and are entirely (ZPAL V.39/34) or almost entirely (ZPAL V.39/66) covered dorsally by the caudal plate of ischium. In larger specimens, the variation is expressed in several ways. Firstly, the caudal processes may be relatively short and rounded distally (CSMM uncat., Figs. 3A and 9A; SMNS 17561, Figs. 3F and 9C; ZPAL V.39/69, Fig. 9F) or relatively long and spiky (SMNS 12777, Figs. 3C and 9B; ZPAL V.39/48, Figs. 5B and 9G; ZPAL V.39/49, Figs. 5C and 9D; ZPAL V.39/56, Figs. 9L–9N; ZPAL V.39/68, Fig. 9H; ZPAL V.39/70, Fig. 9I; ZPAL V.39/71, Fig. 9E; ZPAL V.39/199, Figs. 9O–9Q). Secondly, the lateral edges of the caudal processes are generally thinner than the medial edges, but in ZPAL V.39/56 (Fig. 9N) and SMNS 12777 (apparently—the process is not preserved, but it left an imprint in the rock matrix; Figs. 3C and 9B) the lateral edge is sharpened, while in the remaining specimens it is more rounded. In some cases, this may be an artifact of preservation (the edges are frequently damaged), but some well-preserved and seemingly undamaged specimens (most notably ZPAL V.39/68, Fig. 9H, and ZPAL V.39/199, Fig. 9Q) show that the edge indeed was rounded in some individuals. Thirdly, in some large specimens (ZPAL V.39/69, Fig. 9F; ZPAL V.39/70, Fig. 9I; ZPAL V.39/71, Fig. 9E; Proterochersis robusta specimens are not prepared sufficiently to validate whether this is the case) the ischium is visible in ventral view between the caudal processes, but in most specimens it is not exposed. The degree of this exposure varies from minor (ZPAL V.39/70) to major (ZPAL V.39/69, ZPAL V.39/71). Fourthly, the angle between the caudal processes varies—for example, in ZPAL V.39/68 (Fig. 9H) it is low, in ZPAL V.39/49 (Figs. 5C and 9D) and ZPAL V.39/70 (Fig. 9I) it is intermediate, and in ZPAL V.39/48 (Figs. 5B and 9G) and ZPAL V.39/69 (Fig. 9F) it is larger. Finally, the size and the proportions (length to width) of the intercaudal scute is varied—for example, in ZPAL V.39/68 (Fig. 9H) it is very elongated craniocaudally (2.2 cm in length × 1.2 cm in width), in ZPAL V.39/69 (Fig. 9F) it is nearly as wide as long (2.3 cm in length × 2.2 cm in width), while in SMNS 56606 (Fig. 3J) it seems to be wider than long (but unfortunately damaged). Because there is no clear correlation between these morphologies and size of the specimen, we decided to use shape analysis in search of possible sexual dimorphism (see the section shape analysis below, Figs. 9C–9F).

Based on the width of the caudal plastral lobe, Proterochersis robusta specimen SMNS 17561 (Figs. 3F and 9C) appears to be of roughly comparable size to Proterochersis porebensis specimen ZPAL V.39/66 (Fig. 9J), but the former has a well-developed, adult-like shell (e.g., no middorsal troughs and pronounced keel, cranially protruding gulars and extragulars, caudally protruding caudal processes), while the latter appears to be a juvenile, more similar to ZPAL V.39/34 (Figs. 5A and 9K) than to larger specimens, suggesting that Proterochersis robusta achieved adult-like features (and, supposedly, sexual maturity) at smaller sizes than Proterochersis porebensis. This is congruent with larger average and maximal sizes of Proterochersis porebensis specimens found thus far compared to Proterochersis robusta specimens.

ZPAL V.39/200 (Figs. 9R–9T) is a curious, thorn-like element with lamellar sutural surface at its base. It may represent an isolated caudal plastral process, although it is small compared to other specimens. Otherwise, it may be interpreted as a part of a complex cervical or caudal osteoderm, similar to those described by Gaffney (1990) for Proganochelys quenstedti Baur, 1887, and suggested by Lucas, Heckert & Hunt (2000) and Joyce et al. (2009) for Chinlechelys tenertesta Joyce et al., 2009 (see Discussion in Szczygielski & Sulej, 2018), but no other evidence of such osteoderms is known from Poręba nor German proterochersid-yielding localities.

Shape analysis

No clustering was seen on the PCA plots of the gular region of the plastron, indicating an intraspecific variability (Fig. 8). The plot of regression scores (a projection of the shape of each specimen) vs log centroid size (linked to size) for the gular cross-sections shows no association between the shape and size (Fig. 8F). On the other hand, in the case of the gular and extragular shape in ventral view, the regression analysis shows shape change with the increasing size (Fig. 8E).

In the PCA plot of the caudal processes and intercaudal scute (Fig. 10A) the principal component 1 (65.3% of total variance) connected to the elongation, slenderness, and spikiness of the caudal process. Increasing scores on PC2 (19.7% of total variance) show widening of the connection between the caudal process and the intercaudal scute. The likely juveniles (ZPAL V.39/34 and ZPAL V.39/66) have short and rounded caudal processes. Then, the specimens ZPAL V.39/69, CSMM, and SMNS 17561 placed around 0 value of PC1 and PC2 show longer and rounded caudal process. The most positive values of PC1 are occupied by specimens with very long and spiky caudal processes. In the second PCA plot (Fig. 10B), the PC3 shape changes are connected to the angle between the caudal process and the intercaudal scute and the specimens are placed similarly to former plot.

To check whether the lengthening and sharpening of the caudal processes results from ontogenetic changes, we run a regression analysis (pooled by species, Fig. 10C) of two Proterochersis species. The results show that 56.9% (p-value < 0.001) of the total shape variability can be explained by size variation. The regression plot shows that specimens similar in size can possess short and rounded caudals (ZPAL V.39/48) or long and spiky (ZPAL V.39/69). The result suggests that these morphotypes of developed caudal processes are not a result of ontogenetic changes.

Figure 10 Results of the Principal Component Analysis of the caudal plastral region.

Principal Component Plots showing (A) the first against second and (B) the first against the third Principal Component of the caudal plastral region. The main shape changes for each PC are represented by wireframes, black wireframe is the average shape, while the red line represent shape changes connected to adequate PCs. Results of (C) regression analysis as a representation of the influence of log-transformed size on shape. Proterochersis robusta is represented by stars, P. porebensis by dots. Representations of scute shapes not to scale.

We run a MANOVA for the group including the specimens with short and rounded caudals (ZPAL V.39/69, SMNS 17561, and CSMM uncat.) and the group containing the individuals with long and spiky caudals (ZPAL V.39/70, ZPAL V.39/71, ZPAL V.39/49, ZPAL V.39/48, and SMNS 12777). The MANOVA showed that they differ significantly in shape (F = 2.48, p-value < 0.01) but not in (centroid) size (F = 3.21, p-value > 0.3; Table S6). Therefore, the difference in shape but not in size indicates that the dimorphism is not ontogenetic. Moreover, here is no significant distinction in the shape of caudal process between species (Table S6). It is possible that the differences are caused by the sexual dimorphism, but so far the sample size is too small to unambiguously confirm that.

Scute sulci and proportions

Like in carapace, the width and depth of sulci increase with the size of the animal. Unlike in carapace, the sulci in plastron never seem to undulate and there is no scute striation. In large specimens, however, the sulci do exhibit some minor irregularities, while in small ones they are usually very straight (compare Figs. 3 and 5). As in modern turtles, there is typically some bilateral asymmetry when it comes to the sulci layout on the plastron—most notably, the scutes in corresponding pairs (humerals, pectorals, abdominals, etc.) differed slightly in length, so the points in which their cranial and caudal sulci meet the midline are shifted slightly cranially or caudally relative to each other (this is best visible in CSMM uncat., Fig. 3A). This shift is random, so there is no clear alteration (i.e., the sulci on one side do not always precede the sulci on the other side).

Plastron thickness

The plastron in proterochersid is not equally thick throughout its length and width. It is thinnest in the bridge area and around midline, and thickest in the lateral sections of femoral scute area, where it forms a bulbous expansion. In ZPAL V.39/34 the femoral is 1.8 cm thick, in ZPAL V.39/48 it is 2.1 cm thick, and in ZPAL V.39/49 it is 2.5 cm thick. The thickest plastron found thus far is ZPAL V.39/157, with femoral scute region 2.7 cm thick in the lateral part and about 0.5 cm thick at the medial cranial section of the anal scute area.

Variation of Shell Features

Scute surface

Radial striation

Radial striation and sinuous sulci, similar to those observed in Proterochersis spp. (e.g., Figs. S8A, S8B and S8E–S8G), are present in numerous Triassic (De Broin et al., 1982; Gaffney, 1990) and Jurassic (De Broin, 1994; Anquetin & Claude, 2008; Anquetin & Joyce, 2014; Anquetin, Püntener & Billon-Bruyat, 2014; Jansen & Klein, 2014; Anquetin, Püntener & Joyce, 2017; Sullivan & Joyce, 2017) turtles. The same superficial characters are also present in K. limendorsa specimen SMNS 17757, including clear striation and growth marks along the cranial and lateral edges of the last vertebral scute, and unlike Proterochersis spp. the sulcus between the fifth and fourth vertebral scute of K. limendorsa is clearly sinuous. In that specimen, these characters are more pronounced in the caudal part of the shell than in the cranial part, but this may be a preservation artifact. Due to small sample, it is unknown whether this is taxonomically important, or just related to an old ontogenetic age of SMNS 17757. The ecology of early turtles is controversial, but currently no clear adaptive value is apparent for these low, radial ridges, and given their widespread occurrence in early taxa, they are likely plesiomorphic. Moreover, a delicate striation is present on scutes of some turtles, and possibly in older specimens it may leave imprints on underlying bones.

Growth marks

The presence of bow-shaped scute growth marks (e.g., Figs. S8A–S8F) is most typical for the turtle species in which the scutes are not shed (Zangerl, 1969; Alibardi, 2005). This most usually means terrestrial turtles, but there are also some examples of scute shedding terrestrial box-tortoises or non-shedding aquatic emydids (Alibardi, 2005). The non-pathological growth marks in Proterochersis spp. and K. limendorsa are, however, very subtle, and more comparable to those seen in some specimens of scute-shedding aquatic turtles (e.g., Emys orbicularis (Linnaeus, 1758), Mauremys caspica (Gmelin, 1774); T. Szczygielski, 2018, personal observation), therefore it is not clear whether proterochersids shed their scutes. Furthermore, if scute shedding is an adaptation to aquatic environment (e.g., by lowering drag during swimming thanks to smoother shell surface or as a countermeasure against shell rot), it is possible that this mechanism developed with some delay. In such a case, the scute shedding could have been initially a rare (possibly abnormal) phenotypic trait, only subsequently promoted and eventually fixed by the natural selection some time after turtles invaded the aquatic environment, and thus might have been absent in the earliest aquatic testudinates. Conversely, if scute shedding is plesiomorphic for turtles and was repressed as an adaptation to terrestrial life (by thickening the durable, keratinous layer protecting the epidermis and shell bones), the inability of scute shedding might have also been selected and developed over time in more advanced terrestrial turtles. For that reason, the correlation between the scute shedding and life environment may not be strict for the earliest, Triassic turtles. The oldest (and, apparently, the only known) fossilized isolated turtle scutes were described from the Middle Jurassic of the United Kingdom, but they possess a very dense layout of growth marks and were interpreted as removed post-mortem from the turtle carcass rather than shed (Anquetin & Claude, 2008).

Middorsal ridges and keels

A low middorsal ridge (e.g., Figs. S4C–S4E and S4G) was thus far reported among Triassic turtles only in Proganochelys quenstedti in the cranial portions of its vertebral scutes (Gaffney, 1990). A subtle ridge is also present crossing the area of the vertebral scutes (including the fifth vertebral) of K. limendorsa specimen SMNS 17757. Similar low midline ridges are also present in numerous modern turtles (Pritchard, 2008). While the keels alone are relatively common also in young individuals of modern turtles (G. Ferreira, 2018, personal communication; T. Szczygielski, 2018, personal observation), given current data, it seems that deep troughs lateral to the keels are unique to young stages of shell development of Proterochersis spp. Their genesis and relationship to vertebral scutes, however, are problematic. Certainly, they are not caused by any post-mortem, mechanical folding, because they are symmetrical, their morphology is virtually the same in both specimens, and there is no sign of folding or cracking on the ventral surface of ZPAL V.39/2 (Figs. S4A and S4B). It would be tempting to conclude that in life they were covered by normal, unpaired vertebral scutes, and that the troughs were initially filled by dermis, connective tissue, or rudimentary muscles, and subsequently by developing bone of neurals. The walls of the troughs, including the lateral edges of the midline keel, bear, however, the same rough texture as the remaining, scute-covered surfaces of carapacial bones (contrary to some deeper-located or visceral surfaces, see Szczygielski & Sulej, 2018), suggesting that they were lined by keratinous elements as well. On the other hand, the scutes of larger and probably older specimens do not show any remnants of deep troughs in their older parts. This is problematic, because turtle scutes grow from the bottom (Alibardi, 2005) and the heavily cornified and stiff outer scute layers hardly seem to be susceptible to remodeling. This would call out either for local, temporal scute decornification of mid-sections of proterochersid vertebral scutes, as in plastra of breeding male chelonioids (Wibbels, Owens & Rostal, 1991; Wyneken, 2001; Pritchard, 2008), or for scute shedding in proterochersids, at least when young. Even with help of scute shedding, though, filling of the scute-lined troughs with bone still appears tricky, because the shape of the younger, deeper, and less cornified layer of the scute, which would potentially allow some flexibility and space for bone apposition, would still be determined by the older, stiffer, external layer. It seems, nonetheless, possible that uneven thickness of cornifying epidermis (thicker below the flat parts and thinner at the points of penetration into the troughs) would, with every generation of shed scutes, result in shallower troughs, and eventually in their disappearance. Less probable seems the hypothesis that the newer generations of scutes of Proterochersis spp. attained their stiffness some time after the older scutes were shed, and until then they were pliable enough to allow gradual filling of the troughs, or that the troughs are pathological and developed independently in two specimens of similar sizes from the same locality. Possibly, this conundrum will be solved by future histological studies. Until then, these interpretations remain speculative.

Plastron scutes

The scutes of the plastron in proterochersids (but also in Proganochelys quenstedti) generally lacked undulating edges, superficial striation, and growth marks, in which they differ from the carapacial scutes. In agreement with this, the molecular background of plastral and carapacial scute development is divergent (Cherepanov, 1989; Moustakas-Verho et al., 2014; Moustakas-Verho & Cherepanov, 2015). These differing characteristics may result from different evolutionary history of plastron and carapace—they developed separately and the former appeared before the latter (Li et al., 2008; Schoch & Sues, 2015, 2017)—and possibly are rooted in varied (primaxial vs abaxial) environment of morphogenesis (Burke, 1989; Nowicki & Burke, 2000; Burke & Nowicki, 2003; Shearman & Burke, 2009).

Scute anomalies and growth

Anomalies in scute layout, shape, and number are relatively frequent in turtles (Parker, 1901; Coker, 1905, 1910; Grant, 1936b; Młynarski, 1956; Zangerl & Johnson, 1957; Zangerl, 1969; Cherepanov, 2006, 2014, 2015; Farke & Distler, 2015; Sullivan & Joyce, 2017; Lichtig & Lucas, 2017). Thus far, however, no unambiguous scute anomalies were reported in the Triassic turtles. The only possible exception is the missing first marginal in one of Proganochelys quenstedti specimens but interpretation of that case is uncertain (Gaffney, 1990; Szczygielski, 2017). Some of the morphologies described here were previously noted by Karl & Tichy (2000), but not considered in wider populational or developmental context, but rather glanced over as a part of normal intraspecific variation of their “Murrhardtia staeschei.” The data on Proterochersis spp. presented here reveals, therefore, the first uncontroversial evidence of scute abnormalities in the Triassic turtle taxa.

The most obvious cases concern additional scutes and improper scute shape. The asymmetry of the lateral parts of the first vertebral scute in ZPAL V.39/49, as well as the asymmetrical intervertebral sulci of Proterochersis robusta specimens CSMM uncat. and SMNS 17561, and Proterochersis porebensis specimens ZPAL V.39/34 and ZPAL V.39/72 are minor and may be easily explained as effects of uneven tempo of scute growth in its contralateral parts, and most likely are not caused by improper development (skipped segment or additional placode in vacant myoseptum) and fusion of scute placodes (Cherepanov, 1989, 2006, 2014, 2015; Moustakas-Verho et al., 2014; Moustakas-Verho & Cherepanov, 2015; Moustakas-Verho, Cebra-Thomas & Gilbert, 2017). Likewise, the medial separation of the first pair of abdominal scutes in Proterochersis robusta specimen SMNS 17561 appears to be caused by simple overgrowth of the preceding and the succeeding pair of scutes. As noted by, for example, Zangerl & Johnson (1957), abnormalities in the abdominals are relatively frequent. The supernumerary prehumeral scutes of Proterochersis porebensis specimen ZPAL V.39/385 are, on the other hand, a textbook example of an effect of additional pair of scute placodes. This specimen is even more interesting due to relative rarity of additional scutes in the plastron in many taxa (Newman, 1906b; Lynn, 1937), and rarity of bilateral additional scutes in general (Newman, 1906b; Coker, 1910; Młynarski, 1956; Moustakas-Verho & Cherepanov, 2015).

Proterochersis robusta specimen SMNS 17930 differs from all the other Proterochersis spp. specimens in its abnormally deep growth lines which form “false sulci” along the cranial limits of vertebral scutes. They are well-visible on pleural and vertebral scute areas, and on the latter they are bilaterally symmetrical, which excludes trauma or post-mortem damage from the list of potential causes. Renal hyperparathyroidism (improper bone formation caused by calcium deficiency related to impeded vitamin D metabolism) was reported to cause accentuated “interplates” and growth lines (Frye, 1994; Rothschild, Schultze & Pellegrini, 2013). The morphology of SMNS 17930 may therefore be tentatively interpreted as resulting from this condition, but further studies will validate this diagnosis. Alternatively, if proterochersids did normally shed their scutes, the morphology of SMNS 17930 may be an effect of dysecdysis (scute retention). In any case, this specimen informs about the growth of scutes in Proterochersis spp.

In modern turtles, newly cornified layers of the shell develop below the old scutes (Alibardi, 2005). The scutes typically do not grow with equal speed in all directions, resulting in an off-center position of the oldest (embryonic) part relatively to whole scute area (Zangerl, 1969; Cherepanov, 2015). Based on the growth rings on the scute areas of Proterochersis spp. (including very clear abnormal growth marks of SMNS 17930) and K. limendorsa, it can be inferred that in proterochersids this growth characteristics were also present. The vertebral scutes grew fastest cranially, moderately fast laterally, and slowest caudally. This is also true for the odd-shaped first vertebral, which apparently grew mainly cranially, while its caudal process retained throughout life the same general shape and size. The pleurals grew fastest cranially and lateroventrally, and their oldest areas were likely located close to (or on) the caudodorsomedial bosses. Based on the layout of striation (Gaffney, 1990), vertebrals and pleurals of Proganochelys quenstedti also grew predominantly cranially (not in a radially symmetrical way, contra Cherepanov, 2015). No growth marks are visible on supramarginals and marginals, but the striation of the first and the third supramarginal suggests that the embryonic areas were located, respectively, in the caudal and in the cranial region of the scute area (possibly slightly above the midline). Little can be said about the plastral bones, with the exception of gulars and extragulars, which apparently grew faster dorsally than ventrally, resulting in their ventral curving (see above).

In the light of the above, the deep, sulci-like grooves in the cranial parts of vertebral scutes of Proterochersis robusta specimen CSMM uncat. and shallower, but similar positionally and morphologically depressions of SMNS 17930 and (even less pronounced) SMNS 17561 (Figs. 2F, 2G and 6) are best interpreted as appearing late in ontogeny. As stated above (see Results section), there is a positive correlation between the severity of this morphology and size of the specimen. It is highly unlikely that virtually the same (albeit pronounced with various severity) morphologies appeared ideally medially on several vertebrals of three individuals as a result of trauma or post-mortem damage. For that reason, we interpret this as a developmentally driven scute splitting. Occurrences of splitting (i.e., partially divided) scutes were reported in modern turtles (Parker, 1901; Coker, 1910; Grant, 1936b; Zangerl & Johnson, 1957) and the split usually occurs in the youngest parts of the affected scutes (Coker, 1910; Grant, 1936a; Zangerl & Johnson, 1957). In some cases, this phenomenon may be explained as a result of damage to the epidermis between the scutes, which leads to scar formation and loss of proper cornification ability. In many cases, however, the splitting occurs in regions of asymmetry apparently caused, for example, by pairing of vertebral scute primordia, originating from asymmetrically located scute placodes or just in the middle of a vertebral, and the split divides the areas of particular placodes (Coker, 1910; Grant, 1936b; see Cherepanov, 1989, 2006, 2014, 2015; Moustakas-Verho et al., 2014; Moustakas-Verho & Cherepanov, 2015; Moustakas-Verho, Cebra-Thomas & Gilbert, 2017). Therefore, it seems that in some cases during postnatal life, due to unknown factors, the primarily fused scute placodes may lose connection and start to produce separate scutes. It is, nonetheless, possible, that the lateral integration between the vertebral placodes was relatively weak in early turtles such as Proterochersis robusta (it may be hypothesized that huge size of vertebrals may be partially responsible, for example, by causing some signaling difficulties in large specimens; a mid-portion loss of coordination of cornification front may be also responsible for the asymmetry of the sulcus between the third and the fourth vertebral scute in CSMM uncat.), and for that reason the medial split was relatively common. Curiously, there is no sign of cranial vertebral scute splitting in any specimen of Proterochersis porebensis or K. limendorsa. The sample is nonetheless too small to reliably decide whether the cranial scute splitting in large specimens may be treated as typical for Proterochersis robusta due to some specific developmental, structural or ecological susceptibility. The interpretation of this morphology as abnormality, part of a normal intraspecific variability, or a specific character is, therefore, impossible.

Discussion

Ontogeny

The shell of proterochersids changed during ontogeny in several ways. The most notable is the expansion of gular, extragular, and caudal processes (Figs. 3, 5, 8–10; Fig. S7), and projections of cranial (Fig. 4) and caudal marginals (Fig. S5). In young specimens these elements were short and lack pronounced tips (e.g., Figs. 4A, 5A, 8, 9J, 9K, 10; Figs. S5BE and S5G), while in older specimens they were becoming proportionally larger. The nuchal notch formed by expansion of cranial marginals relative to cervical scute (Fig. 4). Furthermore, the depth and width of shell sulci increased with ontogenetic age (e.g., Figs. S1B, S1L, S1N, S1O, S3, S4 A, S4C and S4E). This is linked to increase of sulci undulation and radial striation of scute areas (e.g., Figs. S8A, S8B and S8E–G). In some subadult and adult specimens, bow-shaped growth marks appeared on the scutes (e.g., Fig. 6; Figs. S8A–S8F). As discussed above, a middorsal keel (e.g., Fig. S4) was apparently present in juveniles and reduced or lost in older specimens. A middorsal keel is also present in juveniles of at least some extant pleurodires (Ferreira, 2018, personal communication) and cryptodires (e.g., Staurotypinae, Testudinidae, Emydidae, Platysternidae, Cheloniidae; Geoemydidae; T. Szczygielski, 2018, personal observation; E. Ascarrunz, 2018, personal communication) but this feature is not documented well in the literature. The dorsal vertebrae were ankylosed in subadults and adults, and the axial skeleton mostly expanded at the points of rib head attachments (e.g. Figs. S1F and S1G). The neural canal changed its shape from vertically elongated to oval (Fig. S2).

Sexual dimorphism

A wide array of sexually dimorphic characters is known in turtles (see Table S7 for examples). Many of these characters, unfortunately, are either unavailable for study in fossil material (cloacal position, hindlimb callosities) or impossible to check in currently collected Proterochersis spp. material due to its incompleteness (e.g., tail length, paw morphology) or damage and distortion (shape of the shell). Among the Triassic taxa, probable dimorphism was proposed for Proganochelys quenstedti in form of two morphotypes of hypoischium (Gaffney, 1990).

As noted above (see Results section), there is a very wide spectrum of sizes, within which the shells of Proterochersis spp. are ankylosed. It appears that all of the individuals possessed a ventral plastral concavity, and while it is possible that the dimorphism was expressed in varied depth or area of this concavity, the breakage and possible compaction of specimens prevent from using sensitive numerical methods, such as the PCA, to reliably check this. While size dimorphism may be present in Proterochersis spp., this difference in size does not explain the presence of ankylosis in even smaller specimens, such as ZPAL V.39/34. Similarly, the incompleteness, crushing, and possible compaction preclude utility of these methods to identify subtle dimorphisms in the carapace. The gular and extragular processes of the cranial plastral lobe and the caudal processes of the caudal plastral lobe, on the other hand, are relatively well-preserved and apparently not deformed in numerous specimens, and these regions of the shell are known to be dimorphic in some Testudines (Brophy, 2006; Pritchard, 2008; Cadena, Jaramillo & Bloch, 2013; Leuteritz & Gantz, 2013; Sullivan & Joyce, 2017).

The tubercles in the gular plastral region are usually larger in males of modern turtles, especially those with combat-based mating behaviors (Pritchard, 2008). This makes the variable gular region of the plastron of Proterochersis spp. potentially dimorphic. While initial observations revealed that there are two main post-juvenile morphotypes of the extragular projections independent from the size of the animal—one with the extragulars ventrally flat and roughly parallel to the cranial region of the humeral, and the second with the cranial tip of the extragulars turned ventrally, no clustering indicative of a dimorphism was found in our PCA and regression analyses (Fig. 8). It may be noted that the performed analyses was only focused on the extragular (and, in ventral view, gular) shape in separation from the rest of the plastron. For that reason it might have downplayed or failed to grasp some of the morphological characteristics of that region, such as the angle between the extragular projection and the plane of the rest of the cranial plastral lobe. Exclusion of the humeral scute area, caudal to the gulars and extragulars, was necessary at this time, because many of the studied specimens lack most or all of that part, rendering reliable positioning of the landmarks impossible. New, more complete specimens and, possibly, utilization of 3D morphometrics may, however, yield other results and should be attempted in the future.

The variability of the caudal processes (Figs. 9 and 10) is seemingly much larger in Proterochersis spp. than the variability in the gular region. Typically, the depth, width, and shape of anal notch is correlated with sex (Brophy, 2006; Pritchard, 2008; Cadena, Jaramillo & Bloch, 2013; Leuteritz & Gantz, 2013; Sullivan & Joyce, 2017). Usually, males have deeper notches than females, to facilitate movement of cloaca-supporting tail during copulation (Brophy, 2006; Cadena, Jaramillo & Bloch, 2013; Sullivan & Joyce, 2017), but there are taxa in which females have deeper notches, possibly to facilitate oviposition (Leuteritz & Gantz, 2013). Proterochersids lack typical anal notch, because their xiphiplastra did not reach the caudal edge of the plastron, and three caudal plastral ossifications (possibly homologous to hypoischium, but more data are needed to unambiguously prove or refute this homology) were instead sutured behind them (Szczygielski & Sulej, 2018), but functionally the space between the caudal processes may be compared to anal notch of pleurodires and some cryptodires. The structure of cloaca and penis is, obviously, unknown for the Triassic turtles, but it may be hypothesized that the notch between the caudal processes would bring them analogous benefits. Furthermore, the hypoischium of Triassic turtles, which was located just caudal to the pelvis and plastron, usually had two fingerlike caudal processes (Gaffney, 1990; Sterli, De la Fuente & Rougier, 2007; Li et al., 2008; Szczygielski & Sulej, 2018). This structure at least in some Triassic testudinates was dimorphic and likely played a role during copulation (Gaffney, 1990; Szczygielski & Sulej, 2018). Although no clear-cut clusters were recovered, the shape analysis shows that the specimens with long, spiky, widely spread caudal processes, such as SMNS 12777, ZPAL V.39/48, or ZPAL V.39/49, do not mix with those with short, rounded caudal processes, such as CSMM uncat., SMNS 17561, or ZPAL V.39/69 (consistently with long caudal process of ischium of ZPAL V.39/69, which blocks the space between the caudal processes and would likely get in the way of male tail), which may hint some (possibly sexual-related) selection toward separation of the caudal processes’ shape in adults. This picture is, however, distorted by odd specimens—ZPAL V.39/68 (long caudal processes, but very close together) and ZPALV.39/71 (long, widely spread caudal processes, but the space between them is at least partially blocked by ischium). It must be kept in mind that this is the worst preserved specimen in the tested group and its edges are worn, so the exact extent of its caudal ischial plate and exact length of the caudal processes may be misrepresented. Unfortunately, ZPAL V.39/69 is too fragmentary to include it in the shape analysis.

Conclusions

Proterochersis robusta and Proterochersis porebensis are found to be very similar to each other in most aspects of their anatomy and variability, although they exhibit some apparently consistent differentiating characters that support their specific identity—most notably, the shape of the caudal notch (semicircular in Proterochersis robusta vs triangular in Proterochersis porebensis; Szczygielski & Sulej, 2016) and the position of the costoperipheral suture in the bridge region (around the middle of the supramarginals in Proterochersis robusta vs at the level of marginosupramarginal sulcus in Proterochersis porebensis; Szczygielski & Sulej, 2018). Here, we also add the maximal size and the tempo of growth (larger in Proterochersis porebensis than in Proterochersis robusta) to this list. Some other characters described here (e.g., tendency for vertebral scute splitting in Proterochersis robusta, more exaggerated morphology of the extragulars in Proterochersis porebensis) may also prove taxonomically diagnostic, but more specimens must be recovered and studied before this is confirmed. Szczygielski & Sulej (2016) also noted the shape of the anterior marginals (serrated in Proterochersis robusta vs straight in Proterochersis porebensis) and the profile of the shell (higher in Proterochersis robusta than in Proterochersis porebensis) as diagnostic for the species. These characters remain in force, but must be treated with adequate awareness. The anterior edge of the carapace is preserved intact only in several specimens, so variability in that regard cannot be refuted, especially in light of the significant variability of marginals described in this work. The exact geometry of the shell, on the other hand, is difficult to establish in Proterochersis porebensis due to crushing or incompleteness of all available specimens. Future discoveries of new material will likely validate or refute the status of these characters.

The observations of shell variation in Proterochersidae reveals, as could be anticipated, that these oldest true turtles being the sister group to all other known testudinatans (Szczygielski & Sulej, 2016, 2018), exhibited numerous progressive characters already resembling derived turtles (e.g., patterns of scute growth, intervertebral position of dorsal ribs), but some either plesiomorphic for Testudinata (radial striation of carapacial scutes) or difficult to assess (deep troughs surrounding laterally the middorsal keel in young individuals, seemingly common medial splitting of vertebral scutes).The presence of growth marks and shell abnormalities comparable to those occurring in modern turtles suggest that the scute system of Proterochersis spp. was already controlled by similar developmental mechanism as in crown group taxa. The cranialmost and caudalmost regions of the plastron are hypothesized to be sexually dimorphic in Proterochersis spp. No clustering was found on the PCA analysis of gular and extragular tubercles, indicating a population variability without a clear cause (Fig. 8). Nonetheless, some indistinct separation can be observed between the specimens with spiky and rounded caudal processes and implies that they may represent a form of sexual dimorphism of Proterochersis spp. (Figs. 10A–10C). The variability in that region is not explained by ontogeny alone. Therefore, it seems that the sexual dimorphism in the anal region of the plastron, which can be observed in some species of extant turtles (most notably pleurodires, but also some cryptodires; see, for example, Brophy, 2006; Pritchard, 2008; Cadena, Jaramillo & Bloch, 2013; Leuteritz & Gantz, 2013; Sullivan & Joyce, 2017) might have been present also in the beginning of their evolution, in the oldest true turtles, around 215 million years ago. Confirmation or refusal of this hypothesis, however, requires a larger sample size and for now this observation must be treated as a pointer for future research rather than definitive statement.

Supplemental Information

Supplemental Information 1 Supplementary information.

Material, extended methods, supplementary tables, supplementary figures, and supplementary references.

Click here for additional data file.

Supplemental Information 2 Models of anterior regions of the turtle shells of Proterochersis porebensis, with sectioned extragulars.

The numbers of specimens are from top to bottom ZPAL V.39/34, ZPAL V.39/48, ZPAL V.39/49, ZPAL V.39/187, ZPAL V.39/333, ZPAL V.39/379, ZPAL V.39/385, ZPAL V.39/387, ZPAL V.39/388, and ZPAL V.39/420.

Click here for additional data file.

Supplemental Information 3 Models of anterior regions of the turtle shells of Proterochersis porebensis, with sectioned extragulars.

The numbers of specimens are from top to bottom ZPAL V.39/34, ZPAL V.39/48, ZPAL V.39/49, ZPAL V.39/187, ZPAL V.39/333, ZPAL V.39/379, ZPAL V.39/385, ZPAL V.39/387, ZPAL V.39/388, and ZPAL V.39/420.

Click here for additional data file.

We thank Rainer Schoch for access to SMNS collection, Carl Schweizer for granting access to CSMM collection, Łucja Fostowicz-Frelik for the macrophotographs of ZPAL V.39/381 and ZPAL V.39/384, Marco Marzola for 3D models of ZPAL V.39/34 and ZPAL V.39/48, Tomasz Sulej for discussion, and Piotr Bajdek for additional preparation work, as well as Jérémy Anquetin (the editor), Eduardo Ascarrunz, Gabriel Ferreira, and Julien Claude (the reviewers), and the production office for their very helpful comments and work needed to process and improve this manuscript.

Institutional Abbreviations

CSMM Carl-Schweizer-Museum, Murrhardt, Germany

SMNS Staatliches Museum für Naturkunde, Stuttgart, Germany

ZPAL Roman Kozłowski Institute of Paleobiology, Polish Academy of Sciences, Warsaw, Poland

Additional Information and Declarations

Competing Interests

Author Contributions

Data Availability

The authors declare that they have no competing interests.

Tomasz Szczygielski conceived and designed the experiments, analyzed the data, contributed reagents/materials/analysis tools, prepared figures and/or tables, authored or reviewed drafts of the paper, approved the final draft.

Justyna Słowiak conceived and designed the experiments, performed the experiments, analyzed the data, contributed reagents/materials/analysis tools, prepared figures and/or tables, prepared 3D models.

Dawid Dróżdż performed the experiments, contributed reagents/materials/analysis tools, prepared and managed 3D models, helped with the Supplementary Materials.

The following information was supplied regarding data availability:

The raw data are provided in the Supplemental Files. The 3D models are available at https://www.morphosource.org/Detail/ProjectDetail/Show/project_id/638.

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
