# Peer review of "Shell variability in the stem turtles Proterochersis spp"

_PeerJ, doi:10.7717/peerj.6134_

## Round 0.1 · original submission · Major Revisions

Three reviewers provided very detailed reports on your manuscript and suggested both minor and major corrections. Please consider their comments with great care when reviewing your manuscript, notably the comments relating to the methods and shape analysis.

·

Basic reporting

The manuscript is clearly written. The English is for the most part correct, and professional. Minor unidiomatic constructions and small grammatical mistakes are scattered over the text but do not compromise its comprehensibility. I have marked and suggested corrections of many instances of such problems in the PDF file. These are most frequently omissions of the definite article (i.e. "the").

I do not have deep knowledge of the literature on turtle developmental biology or detailed stem-Testudines anatomy. To the extent that I could appreciate it, the background information given by the authors was pertinent, sufficient, and provided with a wealth of references, among which were the papers that I would have expected to see cited.

The overall structure of the article is standard and logical. However, I found that the methods section is insufficient for following the presentation of the statistical results. The manuscript would be much improved by moving the text in p. 7 of S1 to the methods section. The figures are very good; clear and detailed. The manuscript makes reference to several measurements of length, width, &c, that are not given in raw form in the supplementary materials. The information is mostly given in the text itself, but if the authors have tabulated that data, it would be very desirable to make it available in a plain text format such as CSV. I appreciate that the authors do provide raw data of the gular region in the form of 3D objects embedded in a PDF file. However, I think it is better to give this data in PLY or STL format, as these formats are open standards and readily readable by most 3D software. 3D PDF was designed for visualisation only, and it makes it very hard if not impossible for other researchers to recover the geometry data of the models.

The manuscript contains results that are relevant to the issues raised in the introduction. Some of the hypotheses (e.g. sexual dimorphism) could have been stated more saliently.

Experimental design

The manuscript fits the aims and scope of PeerJ, as it contains original research in the Biological Sciences that consisted of the collection of very detailed data, and a sound discussion thereof.

The research questions are presented in the Introduction section. The authors explain how their study represents a systematic examination of phenotype variations that are known to be widespread in Recent turtles, but that has been explored only in a few early testudinatans (in good part due to limited fossil evidence, an issue in this study as well).

I found that the authors gave a very attentive and exemplarily detailed description of the shell variation of the concerned species, both in text and figure form. Some parts of the statistical analyses I could not understand. That might reflect problems in the writing of the manuscript, the use of methods unknown to me (in which case I'd appreciate a reference), or the application of the methods themselves. To the degree that I could evaluate it, I found no evidence of tampering with the data or the analyses. I see no reason to doubt that the authors proceeded in accordance to a high ethical standard.

As mentioned before, I think that the methods section could be much improved by adding the text that had been relegated to the supplementary material, and I think further clarifications would still be necessary. For instance, the regression models are not given out explicitly, and therefore they cannot be easily replicated and interpreted.

Validity of the findings

The authors clearly discuss the novelty and relevance of their data in relation to the evolution of developmental processes of the shell in Testudinata, sexual dimorphism, and (inconclusively) to palaeoecological environment. They also suggest avenues of further research in these topics, by means of palaeohistological studies.

The textual description and graphic representation of the data is of high quality, and I expect that the quantitative measurements were acquired with similar care. The practical sample sizes are very small, as it is too common in vertebrate palaeontology. This is of course no intrinsic failing from the authors, but the statistical conclusions that can be derived from the quantitative data should be regarded with extra caution.

The conclusions are clearly stated, responsive to the issues raised in the introduction, and follow from the observed data, although I think they should be tempered slightly in relation to the sexual dimorphism. I have certain doubts and critiques of the statistical methods used by the authors. I recommend major revisions solely because I could not understand all the methods and therefore cannot assess their validity. I have no expectation that the revisions should invalidate the conclusions of the authors.

Additional comments

This article is a valuable contribution to turtle palaeontology, providing data that is useful for the study of character evolution and the inference of the phylogeny of Testudines itself. I stress the latter, because failure to assess phenotypic variability and understand its nature can have grave effects on phylogenetic analyses.

The conclusions reached by the authors seem to me reasonable and consistent with the data. However, the degree of support that is afforded by the data is unclear to me in some cases. Maybe due to miscommunications or actual methodological problems. In either case, I think the manuscript be improved without much extra work.

Starting with the "Methods" section, as I mentioned before, I think it is insufficient for a self-contained reading the manuscript. Moving the "Principal Component Analyses" section in the supplementary materials to the "Methods" section of the main article would go a long way in making the article more comprehensible (by the way, the "Principal Component Analyses" section covers PCA only in its last paragraph, most of the text is actually about data acquisition and GPA superposition). Further, the version numbers of all the software used should be given, as this allows more faithful replication. Different versions can have different feature sets, algorithmic implementations, and bugs. In a more technical and admittedly pedantic note, MeshLab is distributed as "free software" under a GPL 3.0 Licence, which is different to what is commonly meant by "freeware". The concept of free software is based on the idea of freely sharing the source code, while freeware can be any software whose binaries are distributed at no cost, but the source code might remain undisclosed.

Moving on to the results, there are many references to "correlation indices" between linear measurements. I am not familiar with this term, the authors probably meant "correlation coefficient" (eg., p.21.l.462, p.25.l.601). The type of correlation coefficient (probably Pearson's) should also be reported, as well as the associated p-value. Those correlation coefficients were obtained from only 4 observations, so I think the authors should be more cautious about the strength of the evidence for the length correlations. Consider that n=3 is the bare minimum number of observations necessary for estimating the fit of the data to a linear model.

In p.30.l.735-44 a regression analysis is said to have been used to determine the presence of separate clusters in the shape of the ventral view of the gular region. I do not understand what regression model was used. What were the dependent and independent variables? What was the regression coefficient? More importantly, I do not understand how regression analysis was used to determine cluster separation. An explanation of the logic of this procedure and maybe a reference seem necessary.

In p.34.l.872-75 the authors claim that the PCA of the caudal region shows two shape clusters (and two singletons corresponding to juveniles). These clusters are not so apparent to me by the naked eye, certainly much less than the clusters of shapes of the vertical cross-section of the extragulars. For instance, in the PC1-PC2 plot SMNS 17561, which is assigned to "Group I", is practically equidistant to V39/69 in "Group I" and SMNS 12777 in "Group II". "Group II" could also be arguably divided into the shapes with PC2 > 0 and the shapes with PC < 0. The cannonical variates analysis that the authors perform does seem to validate the separation of the two clusters (with a rather large p-value, bearing in mind that the significance threshold of 0.05 is an arbitrary convention), so they might reflect some relevant biological phenomenon, but this still does not explain to me how the clusters were identified in the first place. An implicit and reasonable assumption seems to be that the number of clusters must be two in order to match the sexes, but no further criterion is apparent.

A way to validate "Group I" and "Group II" or to find alternative groupings is to use a partitional clustering algorithm such as k-medoids or k-means, among many others. These algorithms are not infallible, and might not yield optimal clusters for instance when the clusters have very different variances. Should the authors want to use them, they ought to check that the limitations of these algorithms and how they relate to their assumptions and the nature of their data. k-medoids and k-means are available in multiple R packages, and k-means is available in the software PAST. k-means is known to be more sensitive to outliers.

Alternatively, if the authors had an a-priori expectation of the shapes that would be clustering into "Group I" and "Group 2", that ought to be explained.

The CVA used to validate the clusters of the caudal region can also be used for the clusters of the vertical cross-section of the extragulars. It is not clear to me why the authors did not employ this method in the two cases.

Finally, in p.35.l.896-98 the authors mention a regression analysis, but again it is not clear on what variables the analysis was performed. Fig. 10C shows that the centroid size is the independent variable, but the dependent variable is given as a "regression score" of meaning unknown to me. Further, I do not know if a log-transformation was employed, as it is customary in allometric models. This analysis merits a more detailed explanation.

I provide the annotated review PDF so that the authors can check the language problems that I spotted.

·

Basic reporting

The manuscript is of very good quality and the findings are important to the understanding of the early evolution of turtles, showing that some of the common intraspecific variability seen in extant turtles were already present in the earliest representatives of the group. Although none of the authors is a native English speaker (but neither am I), the manuscript is well-written and the language used throughout the text is adequate. I made some specific corrections along the manuscript (please find all of those in the annotated PDF) and some suggestions on the usage of specific words, more specifically some anatomical orientational terms, that, in their current usage, make the manuscript sometimes difficult to follow.
The Background and Introduction sections show the relevant context for the development of this research and Literature is well referenced all along the manuscript. The structure of the manuscript conforms to PeerJ standards. I felt that the Results section, especially the descriptions of variability, is a bit too extensive and could be summarized to make the text more fluent and easier to read. It is difficult for me to tell what is more relevant and what could be may be sent to the supplementary files, but do consider revising that.
Figures (as well as supplementary figures) are relevant, high quality, well labeled and described. I suggested the inclusion of one additional figure (that could be added to the supplementary file without any problems) to more clearly refer to some specific variation on shell scutes.
The raw data in itself seems alright to me, but I consider it problematic that the authors didn't fully present some of them, e.g., complete descriptions (with reference to figures) of the sets of landmarks used for the shape analyses, and that the description of some analyses (especially the correlations/regressions) are only very briefly state in the supplementary files.
Please, find below more extensive details on these issues. In conclusion, I support the publication of this manuscript after revision of those points.

Experimental design

The research is original, with well-defined, relevant and meaningful questions presented and answered by the authors and the experimental design is adequate to this study.
The methods, although adequate, are not fully described in the manuscript, hampering replicability of their findings. Even though it appears to me that all analyses were conducted properly, the methods are described only partially in the supplementary files. I strongly feel that it is necessary to describe in more detail some of the analyses the authors conducted (especially correlation/regression analyses, and the landmarks used in each set of analyses) and I advise to move at least a brief description of the conducted quantitative methods to the main text (Methods section), instead of leaving everything in the supplementary files, since those are a very important piece of this research.
Also, I felt that the manuscript would benefit if the authors created a "Shape Analyses" section in the Results. Describing all morphological variation at first and then presenting the results of quantitative analyses to back some of the generated hypotheses seem to me a more straightforward and easy-to-follow way to present their results. Even if you do not agree to make a new section, the results of the quantitative analyses should be described in more detail. More specifically when the authors report and discuss the variations along the Principal Component and Canonical Variation axes, it is critical to report what do these variations mean in morphological terms, otherwise, it is just a graph. So, for example, when saying that Groups I and II are separated along the PC1 axis, what do the variation on positive and negative values of this axis mean for the morphology of the analyzed element?
If the descriptions and report of the methods, I cannot consider the methods description sufficient for replicate.

Validity of the findings

The manuscript presents important new data, with some novel interesting insights about the morphological evolution of early turtles. The results are described in detail, proper analyses are conducted (although I suggested they should be better described) and the discussion and conclusions, even though I made few suggestions on specific parts, are well stated and adequate to their findings.

Additional comments

I made several minor and some major comments along the annotated PDF of the main text and the supplementary file S1 as well, please, do consider those when revising the manuscript.
I consider it important to review the usage of some terms, such as cranial/caudal and anterior/posterior, and 'ontogenetic age' all along the manuscript. In the case of using cranial/caudal and anterior/posterior there is no agreement on which set of terms are more adequate and you can choose any of these, but, you should use those consistently through the text. When you use both cranial and anterior in the same text (sometimes in the same paragraph) the reader feels that you want to refer to distinct issues, i.e., that cranial means one thing in this context and anterior another. I do not think that was your goal, so I do advise you to choose and use just one kind of these terms. If that was your goal, you should more explicitly define what you meant by each of those. Other terms you used along the text, such as 'behind', 'higher', etc., I consider ambiguous terms for anatomical descriptions. Please, do consider also reviewing them and changing for more straightforward terms, e.g., posterior to, dorsal to, etc.
Using 'ontogenetic age' in the descriptions also bothered me because you cannot define an age for those materials without using their sizes as proxies. So, why not use size instead? Larger individuals, instead of older individuals, for example, are more appropriate in the description. You can (and should) approach the ontogenetic age in the Discussion section, when you will refer to larger or smaller individuals and discuss their inferred age relations.

·

Basic reporting

The English is fine (at least for me, I am not English speaker)
The references are fine
The article structure could be a bit improved (see remarks about cutting discussion/results differently)
Except potential circularity with statistical methods, the results and interpretation sound to me (see general comment).

Experimental design

The morphometric methods and statistical procedure should be better explained even in the appendix. As it stands, it would be difficult to reproduce easily the same study because there is not enough explanations about procedures.

Validity of the findings

The interpretations sound to me, the conclusion are well supported by the study.

Additional comments

main document:

This is an interesting, long, exhaustive, and complete study about shell variation in Proterochersis. There are few similar studies done on fossil or living turtles and I was very interested by the text.

I found the english understandable and correct (I am not english native); the ms is densely illustrated and the fossil material is remarkably well described.

I have a set of remarks, mostly related to statistics and to few other points that could be considered to improve the ms.

MAJOR REMARKS:

M1* the principal components are extrapolated from the shape of individual specimens, I recommend to display minimum and maximum of shape variation on each PCs since it may better show the meaning of axes in terms of "morphological" variation. The same applies for the CVA.

M2* since the idea is to discover a grouping structure, the canonical analysis may not be appropriate since it is circular reasoning (you based your group on the way they differentiate on PCs, and of course, as a result, they will differentiate morphologically). Rather you might prefer to use gaussian mixture models or kmeans or partition around medoids to identify whether you have several groups or not...I also cowrited a paper some years ago with a Bayesian approach (see geneland or things I wrote in the morphometric book).

M3*The long description of shell morphology and shell variation would gain to be accompagned by a table where you might note which character are overlapping in morphology accross both species, and whether some characters are really specific.

M4*From what I saw in the text and from the analysis, it seems that sex dimorphism encompass largely interspecific variation. Why therefore not being more conclusive and say that these two species differ by so few things (or nearly nothing); note that the characters that diagnose the two taxa are also the more variable and vary importantly during ontogenesis or can be related to sex dimorphism (shape of the carapace margin, anal notch...: see for instance what happens in giant galapagos tortoises, especially in saddleback forms). It is a bit confusing then to poo the two species and to analyse them jointly. Alternatively, you may drop from your analysis things related with specific variation and concentrate only on one species.

M5*There are many things in the discussion that could appear before with the section results (which could be renamed "Variation of shell features"). I would rather concentrate the discussion on sex dimorphism and ontogenetic variation, and possibly interspecific differences. The other sections (middorsal ridges and keels, growth marks, radial striation) within the previous chapter (it is more a description than a discussion here).

M6*shape analysis:
- L. 742. It is unclear on what is regressed shape ? is it regressed on size, and then after is it a pca on residual variation ? What is the p-value refering to ? to the significance of the regression on size ? If so it seems that shape variation is not related to ontogeny ?
How sex were a priori defined on this plot if it is not by the shape of the structure ?
What about the two groups on PC3 for the vertical section ? why PC1 is interpreted as sex related but not PC3? I think it would be necessary to be more precise here because conclusion of the morphometric analysis are a bit tautological.

- fig 10.C, it is unclear what the authors mean by regression score, the way to obtain this score should be explained in the text. The construction of groups look rather ad-hoc, for instance smns17561 could be in group I or II, and the same could apply to V39/70. Here also, using partitional clustering could have been informative (eg. gaussian mixture models, kmeans, or other related methods...). Think that PC axes are in no way biological axes, they just represent major component of variation and covariation. Although I think there is clearly an inflation of variable by comparison to the number of individuals, it would have been good to provide what were the shape characters related to group differentiation by reconstructing extreme shapes along CVA axes. As for the CVA, the manova run on the groups was expected: it is a circular result (if you define the group on the based on their shape variation, you should expect to find significant differences in shape). I am not sure that these analyses are necessary. The CVA could be interpreted, if you compute mathematically the shape diagnostic features among groups.

M7* The methods used for geometric morphometrics are not enough explained in supplementary material.
- The location of landmarks and semi landmarks should be described by the mean of a drawing, or by the mean of a list.
- since there are potentially few individuals but large number of variables, it is not very surprising to see clear differences in canonical analyses. Can you explain whether you reduced shape space dimensionality prior to run the canonical analysis (how many PCs axes were kept ?).

OTHER REMARKS:

o1*The chapter method starts a bit abruptly, can you first explain the purpose of taking photograph and how you made photograph for most specimens rather than only discussing about the small.

o2*l. 150 and everywhere in the ms. I am not sure that we speak about ontogenetic age, the meaning of that expression is unclear to me. Maybe rather, you want to say a large spectrum of shapes ? I am not really sure that we speak about ontogenetic stage after hatchling or it requires to be better defined.

o3*l.678-679: rephrase a bit, I was not understanding that sentence.

o4*l. 687. "an earlier stage of hyoplastron formation". should be rephrased, I do not understand it.

o5*l. 176-177. a bit unclear, I would rephrase it. I was thinking that carapace stoped growing, not that ankylosis was.

o6*Ankylosis remark: Your observation is interesting, and I believe that ankylosis is not necessarily related to stop of growth (it may be the case in Terrapene, but not necessarily in other turtles: eg. Batagur, and certainly the case in some fossil such as Meiolanids : bone recomposition is likely to occur even if the suture disappear). Therefore I tend to agree with your conclusion, one should not necessarily conclude that sex maturity is related to ankylosis and that carapace stop growing once ankylosed. In this case, you cannot cross the information of size and ankylosis to determine easily the sex of the turtles, and you may rely on shape (and its covariation with size).

o7*l. 275 to l.300. when you speak about growth in some body part it would be probably more informative to speak about change in proportions; indeed everyone expect that by the simple process of growth, measurements will all increase. (or at least and if it is the case, just state that the growth is isometric).

o8*When you first mention a taxon in the text, please add author and year (you do not need to list authorship of taxa in the bibliography: eg. Keuperotesta limendorsa ).

o9*Fig 8. there is no fig 8F (correct 8D).

o10*l.858. "see the secion shape analysis below" -> as for thue gular part, this section is not clearly identified in the text, it seems to belong to the same chapter.

o11*Discussion:
As I mentioned during the turtle evolutionary congress in Tokyo, the fact that carapace variation is rather important in this group and could be related to the ecology of Proterochersis. Proterochersis was likely more terrestrial than Proganochelys, it would be therefore expected that the anal region would be more impacted (reproducing on land imposes more important constraints for reproduction). Although, authors speak about gular region and possible fight. If they are right in their interpretation, it further suggest that these turtle were terrestrial. Gular fight or strong differentiation was never reported for aquatic species. Maybe that should appear in the discussion.

o12*supplementary material
p7.l7. sup mat 1, veritcal -> vertical
interscutal -> not sure that this word is used for turtle anatomy. It seems more devoted to acaralogy
extragularium and gularium as well are not commonly used for turtle anatomy

I am julien CLAUDE.
Institut des Sciences de l'Evolution de Montpellier. feel welcome to contact me if you need more info concerning any of my remarks.

---

## Round 0.2 · Minor Revisions

Since there were significant revisions to the first version of your study, I asked the three initial reviewers to provide their opinion on this revised version. They all agree that your paper has been greatly improved, but two of them consider that some important points still merit your attention. Please, read these comments in detail and update the manuscript accordingly.

·

Basic reporting

The changes introduced by the authors significantly improved the readability of the manuscript. It is now easier to follow the relation between some of the methods used by the authors and the interpretations of their results. As before, I added a few suggestions to the text. The authors have now also included their data in a more accessible form,which is very appreciated.

Experimental design

The methods used by the authors are much more clear now. Otherwise, I have nothing to add that wasn't said in the first review.

Validity of the findings

In my opinion the main problem that remains in this second revision is the interpretation of the results of the analyses of the caudal region of the plastron. Although in many cases it is possible to identify clusters in PCAs plot by eye, figure 10 A & B are not one of such cases. I have removed the colour and group ellipses from Fig. 10 (see caudal_pca.gif) and see no clusters. Qualitative identification of clusters is of course subjective; my personal criterion would be that the clusters must be immediately apparent. Going a step further, I scraped the coordinates of PC1, PC2, and PC3 from the plots in Fig. 10 A & B and ran two simple analyses with the partitioning around medoids algorithm implemented in the R package "cluster" v. 2.0.7 (the scraped data and an R script with all the code necessary are provided). In the first analysis I included all the specimens and set the k parameter (number of clusters to find) to 4. In theory, this analysis could have found the same groupings shown in Fig. 10A: group I, group II, and two putative juvenile singletons. The algorithm did not find the clusters suggested by the authors (see clusters_all.png). As this failure to replicate might've been due to the influence of the juvenile specimens, I ran a second analysis omitting V.39/34 and V.39/66 with k set to 2. This other analysis failed to recover group I and group II again. Instead, it found a cluster with V.39/70, V.39.71, and V.39/49 (all with positive values in PC1 and PC2), and another cluster containing all the other specimens.

As I cannot replicate qualitatively or quantitatively the groupings claimed by the authors, I'm afraid that I find their interpretation of the PCA analysis unconvincing and their overall conclusion about sexual dimorphism poorly supported.

Additional comments

Should the you want to use the results of the clustering analysis in their paper, you're welcome to do so, but I must add a note of caution: Partitioning cluster algorithms will always identify the number of clusters required (k) in empirical data, regardless of whether the data contains "real" clusters or not. Several statistical methods exist to assess the support of clusterings; my perception is that there is no clear clustering structure in the caudal region data.

The paper provides valuable new data and insights, and I hope to see it published soon with revisions to the PCA analysis and its interpretation.

·

Basic reporting

The revised version of the manuscript completely and satisfactorily answered all my comments and see no need for further review. Hence, I now recommend the acceptance of the manuscript as it is.

Experimental design

no comment

Validity of the findings

no comment

·

Basic reporting

The authors have answered several of my questions, but the answered raised the problem of a possible circular reasoning in the shape and statistical analysis. This should be taken into account. see the review in section general comments.

Experimental design

ok but see section 1. see the review in section general comments.

Validity of the findings

ok but see section 1 and think that anatomy can also speak, there is no superiority of approach between an anatomical exploration and morphometric and statistical methods. see the review in section general comments.

Additional comments

This MS is still interesting (nice description/comparison of the material) and I am happy to have interacted with the authors during the review process.
The authors have done important changes in their text, they answered most of my questions and the manuscript is much better but I still find things that must be improved in the shape analysis section to avoid circular reasoning; they also asked me to provide a few how to, which is provided. there are a few minor things to correct also with the new changes operated. I consider that the changes I am asking for are minor by comparison to what as been done since the last version, and I thank the authors for taking my previous remarks into consideration.

#Shape analysis and remarks to my previous review: starting from l.1136. points M1,M2, M6/7 (the most important to my opinion), points to be adressed in more details.

M1. We still hardly understand mathematically the relationship between PC axes and shape. What we ideally need is an illustration of what is actually ongoing on PC1 and PC2 or CV1, CV2 (see for instance Claude et al., 2003, Claude et al. 2004...and many other references available for other organisms). basically to obtain that:
for a PCA
1.compute the mean shape
2.estimate max and min scores on PC1, PC2
3. compute change on PC1 by reconstructing hypothetical shape on it for min and max score:
mean shape + eigenvector 1 * max (score PC1)
mean shape + eigenvector 1 * min (score PC1)
the same can be done on PC2
mean shape + eigenvector 2 * max (score PC2)
mean shape + eigenvector 2 * min (score PC2)

Doing so, you may have a clearer idea of what is described by PC1.
for the CVA (something that is equivalent to linear discriminant analysis), the procedure is described with R code here:

http://www.italian-journal-of-mammalogy.it/Log-Shape-Ratios-Procrustes-Superimposition-Elliptic-Fourier-Analysis-Three-Worked,77247,0,2.html

or in my 2008 morphometric book

There is a need here to rescale the discriminant functions since they are expressed in the discriminant space with the mahalanobis distance (not the euclidean one)

M2. the absence or presence of clustering on a PC plot does not mean that the data are actually clustering or not (you see just one plan and data here are highly dimensional). In addition clustering by hand or by eye on the first plan is rather subjective. To demonstrate any clustering in terms of shape, you should use partition clustering methods or eventually superimpose a hierarchical clustering onto your acp (upgma for instance). I prefer partition clustering. In the rebuttal letter, you asked the step by step procedure. That's not really difficult if for instance you opt for multivariate gaussian mixture modelling.
1. consider the scores on your first PCs (for instance summarizing up to 90% of variance) to avoid an inflation of variable by comparison to the number of specimen. save that in a table
2. download R and install Mclust library
3. import the table in R, assign it to the data1 object (with read.table)
4. then run mclust: mod<-Mclust(data1)
5. mod, the output will let you know how many groups you have and which gaussian mixture model it conforms.
6. if you want to check the classification, you can extract the object mod$classification which will let you know to which group belongs each observation. do not forget to cite references to Mclust in your text if you use it.

...you might be disappointed by the result if you put too much variables because in that case, Mclust may conclude that you have only one group (multivariate gaussian models may require too much parameters); this is why I advise to reduce the data to their first principal components of variation. you can also force Mclust to build two groups (option G=2) to see what it returns to you at the end (I would exclude the supposed juvenile here since we expect them to be not differentiated ni terms of sex shape dimorphism).

if you do not like Mclust, just have a look in Claude 2008, I provide an how to for kmeans (which is a simpler method aiming at maximising intergroup variation and minimizing intragroup variation). and if it is still not exactly what you are looking for just check what we wrote for geneland in systematic biology with Guillot.

M6/M7. shape analysis and possible circularity in terms of reasoning.

- shape against log size regression analysis.
Thank you for clarifying this point:
Here also, since in the paper you describe ontogenetic trends, it would be good to reconstruct the shape using the regression parameters rather than to display individual shape.
for that your shape X = A log(size) + B
X being your shape expressed as a vertical matrix, A and B being vectors of regresion parameters.
to reconstruct shape variation due to size, you can reconstruct two shape: the one for the smallest and the one for the largest individual in your set. It will show you which shape parameters are described by the multivarite regression.

"- since there are potentially few individuals but large number of variables, it is not very surprising to see clear differences in canonical analyses. Can you explain whether you reduced shape space dimensionality prior to run the canonical analysis (how many PCs axes were kept ?)."

We do not know how to reduce the shape space dimensionality in MorphoJ. If the Reviewer considers that important and provides us with a step-by-step instruction of how to do that, we may include this.

Discrimination, sex determination, MANOVA and circular reasoning.
This is certainly the most important point to address to my opinion.
Thanks to your explanation concerning sex assignment, I know now that you are basing your grouping structure on shape (PC differentiation). Therefore, it is not surprising to find a linear combination of variable discriminating the groups (your answer to my remark about the sex determination is not equivocal since you say that "The sexes are marked based on PCA clusters"). Therefore and indeed, we expect to find a significant p-value for shape variation explained by size in the manova and we expect to find a nice differentiation on the LDA. Since this is tautological, I think you should drop the LDA and MANOVA.

If you opt for keeping the CVA, I think the problem of number of variable contra number of observation is an important point to consider: when you have plenty of variables and few individuals you can easily obtain a discrimination that is just dependent of the sampling and that do not reflect reality. To reduce shape dimensionality (but it may not apply anymore since the sex were obtained from morphometrics), you can concentrate only on the scores of the first X PCs describing your shape and discard the one contributing for a very small percentage. With your data, I would try to gather 90 of shape variation and drop the next PCs. Note that procrustes manova is also biased since the number of degrees of fredom increase with the the number of points.
To come back on the sampling artefact, you must check whether your specimens are well reclassified by performing a one leave out cross validation on your CVA or LDA (it is a jacknife) and it is explained in my 2013 paper in Hystrix (you have a tutorial in the supplementary material of the paper and everything is online).
* * *
#New minor things :
on1. since you explain now how you buid your 3D models, you should provide the reference for agisoft photoscan (not only for meshlab and visualSFM).

on2. reference to morphoJ should be restricted to the paper published in 2011 (the 2009 paper is not specific to that soft).

Julien CLAUDE, ISE-M
Université de Montpellier

---

## Round 0.3 · Minor Revisions

Thank you for showing a great care to take the reviewer's comments into account. I think your manuscript is now ready for publication.

Since I forgot to send you the additional files provided by one of the reviewer during the last round of reviews, you informed me by email that you would like to make a few minor corrections to your manuscript to take these last comments into account. I therefore invite you to submit a revised version of your work, which I will then accept for publication. I apologize again for the mistake.

---

## Round 0.4 · accepted · Accept

Thank you for these last corrections. I am pleased to confirm that your manuscript is now accepted for publication in PeerJ.

#